# The stability of the primed pool of synaptic vesicles and the clamping of spontaneous neurotransmitter release rely on the integrity of the C-terminal half of the SNARE domain of syntaxin-1A

Andrea Salazar Lázaro[1], Thorsten Trimbuch[1], Gülçin Vardar[1]*, Christian Rosenmund[1,2]*

[1]Department of Neurophysiology, Charité-Universitätsmedizin Berlin, Humboldt-Universität zu Berlin, Berlin Institute of Health, Berlin, Germany; [2]NeuroCure Excellence Cluster, Berlin, Germany

*For correspondence:
gulcinv@gmail.com (GV);
christian.rosenmund@charite.
de (CR)

Competing interest: The authors declare that no competing interests exist.

**Abstract** The SNARE proteins are central in membrane fusion and, at the synapse, neurotransmitter release. However, their involvement in the dual regulation of the synchronous release while maintaining a pool of readily releasable vesicles remains unclear. Using a chimeric approach, we performed a systematic analysis of the SNARE domain of STX1A by exchanging the whole SNARE domain or its N- or C-terminus subdomains with those of STX2. We expressed these chimeric constructs in STX1-null hippocampal mouse neurons. Exchanging the C-terminal half of STX1's SNARE domain with that of STX2 resulted in a reduced RRP accompanied by an increased release rate, while inserting the C-terminal half of STX1's SNARE domain into STX2 leads to an enhanced priming and decreased release rate. Additionally, we found that the mechanisms for clamping spontaneous, but not for $Ca^{2+}$-evoked release, are particularly susceptible to changes in specific residues on the outer surface of the C-terminus of the SNARE domain of STX1A. Particularly, mutations of D231 and R232 affected the fusogenicity of the vesicles. We propose that the C-terminal half of the SNARE domain of STX1A plays a crucial role in the stabilization of the RRP as well as in the clamping of spontaneous synaptic vesicle fusion through the regulation of the energetic landscape for fusion, while it also plays a covert role in the speed and efficacy of $Ca^{2+}$-evoked release.

## eLife assessment

This **important** study presents a series of results to uncover the role of C-terminal half of the Syx1 SNARE domain. The evidence supporting the conclusions is **convincing**. The paper will be of broad interest to biophysicists and neurobiologists.

## Introduction

The molecular release machinery underlying synaptic vesicle fusion in mammalian central synapses consists of the SNARE complex at its core and several accessory proteins. The SNARE complex is formed by the interaction of plasma membrane-anchored proteins syntaxin-1 (STX1 refers to syntaxin-1A and syntaxin-1B throughout this study unless stated otherwise) and SNAP-25 and vesicle-associated membrane protein synaptobrevin-2 (VAMP/SYB2). The SNARE proteins assemble their 55-amino-acid-long, highly conserved SNARE motifs into a parallel four-helix bundle. The resulting

SNARE complex that zippers up sequentially from N- to C-terminus physically brings the vesicle and plasma membranes together and provides the necessary energy to mechanically force membrane merger (*Fasshauer et al., 1998*; *Sutton et al., 1998*; *Sørensen et al., 2006*; *Weber et al., 2010*; *Zhang, 2017*; *Zhang et al., 2022*). The assembly of the SNARE complex creates a highly charged outer surface (*Fasshauer et al., 1998*; *Sutton et al., 1998*; *Ruiter et al., 2019*) to which accessory proteins such as complexins, synaptotagmins, or Sec1/Munc18(SM) proteins can bind to and thereby modulate the efficacy and timing of neurotransmitter release (*Rizo, 2022*; *Rizo and Rosenmund, 2008*; *Südhof, 2013*).

Among the unique components of the SNARE complex, STX1 stands out for its complex structural characteristics. On the N-terminal extreme of STX1, the short regulatory N-peptide and the autonomously folded three-helical $H_{abc}$-domain are essential for the formation of fusogenic SNARE complexes and the readily releasable pool (RRP) of vesicles (*Dulubova et al., 2007*; *Fernandez et al., 1998*; *Gerber et al., 2008*; *Khvotchev et al., 2007*; *Ma et al., 2011*; *Vardar et al., 2021*; *Zhou et al., 2013*). On the C-terminal extreme, STX1 contains the juxtamembrane domain (JMD) and the transmembrane domain (TMD) that are actively involved in membrane fusion (*Dhara et al., 2016*; *Gao et al., 2012*; *McNew et al., 1999*; *Sutton et al., 1998*; *Vardar et al., 2022*). The most crucial, however, is its SNARE domain as it directly regulates the membrane merger. SNAP-25 and SYB2 mutation studies that destabilize the inner hydrophobic pocket of the SNARE domain have determined that the SNARE motif of these proteins can be functionally compartmentalized into two subdomains: the N-terminus, necessary for priming and nucleation of the SNARE complex, and the C-terminus, essential for fusion (*Gao et al., 2012*; *Sørensen et al., 2006*; *Walter et al., 2010*; *Weber et al., 2010*). This was further confirmed by single-molecule optical tweezer studies, which determined the sequential assembly of the SNARE complex through diverse stages assisted by these two subdomains (*Gao et al., 2012*). Additionally, it has been determined that accessory proteins, such as Munc18-1 and complexin, chaperone the stability of the N-terminus and facilitate or clamp the assembly of the C-terminus in the partially zippered SNARE domain, which without their regulation is fast and spontaneous (*Gao et al., 2012*; *Hao et al., 2023*; *Ma et al., 2015*). Although studies using STX1/STX3 chimeric constructs revealed that the SNARE motif of STX1 may carry structural features that regulate spontaneous release (*Vardar et al., 2022*), research on the SNARE domain of STX1 has been limited to the studies of its cognate interaction partners such as Munc18-1, synaptotagmin-1 (SYT1), and complexin (*Hao et al., 2023*; *Jiao et al., 2018*; *Ma et al., 2015*; *Zhou et al., 2015*; *Zhou et al., 2017*).

In this study, we address the specific function of the entire SNARE domain of STX1 in regulating neurotransmitter release in intact synapses. For that purpose, we used a comprehensive structure–function approach and compared the efficacy of STX1 to the closely related isoform STX2 (*Sherry et al., 2006*), which is implicated in the secretion of various peripheral secretory cells (*Abonyo et al., 2004*; *Dolai et al., 2018*; *Hutt et al., 2005*). When expressed in STX1-null hippocampal mouse neurons (*Vardar et al., 2016*), STX2 synapses showed impaired vesicle priming, unclamping of spontaneous release, as well as inefficient and slowed Ca²⁺-evoked fusion, consistent with an impaired SNARE complex function (*Hao et al., 2023*; *Jiao et al., 2018*; *Lai et al., 2017*; *Stepien et al., 2022*; *Voleti et al., 2020*). Therefore, we proceeded with a structure–function chimeric analysis between STX1 and STX2 to isolate the precise involvement of the SNARE domain of STX1 in the regulatory mechanisms of release. We found that the C-terminal half of the SNARE domain of STX1 is a key regulator in the stability of the RRP and is part of the clamping mechanism for spontaneous release in central synapses. Additionally, while it may have a role in the regulation of the speed and efficacy of Ca²⁺-evoked release, these functions depend on the integrity of the full SNARE domain and other domains of STX1. Overall, we discovered that the role of the SNARE complex goes beyond zippering and interacting with regulators of release and has evolved to participate in the fine-tuning of the fusion energy barrier and the precision of synaptic vesicle release alongside its accessory proteins.

## Results

### STX2 supports neuronal viability

Previously we have shown that comparison of non-canonical syntaxin isoforms to STX1 can offer insight into the specificity of vesicle fusion (*Vardar et al., 2022*). For this, syntaxin-2 (STX2), syntaxin-3 (STX3), and syntaxin-4 (STX4) are the best candidates as they share the same domain structure with

STX1 (*Quiñones et al., 1999*). For example, the isoform STX2 has been found in brain tissue and is present in the synaptosome fraction (*Chen et al., 1999*). STX1A and STX2 share a 63% total sequence homology and a 69% homology in their SNARE domain (*Figure 1A*). As expected from the promiscuity of syntaxins in particular and SNARE proteins in general, overexpression of STX2 can rescue neuronal survival in the absence of functional STX1 by binding to SNAP-25 (*Peng et al., 2013*), although its cognate SNARE partner in non-neuronal intracellular trafficking is SNAP-23 (*Abonyo et al., 2004*).

Having this in mind, we lentivirally expressed STX2 in STX1-null neurons using our well-established model of STX1-null mouse hippocampal neurons (*Vardar et al., 2016*) and conducted a series of rescue experiments. STX1-null neurons typically do not survive in culture longer than 8 days in vitro (DIV) and exhibit a complete loss of neurotransmitter release (*Vardar et al., 2016*; *Figure 1C*). Yet healthy synapses are vital to provide a solid background to further analyze the electrophysiological properties. Therefore, our first objective was to compare the ability of STX1A and STX2 isoforms to rescue neuronal survival. We quantified the amount of mouse hippocampal neurons in high-density neuronal cultures at DIV8, DIV15, DIV22, and DIV29 for each group and found that STX2 rescues neuronal survival at a comparable level to STX1A (*Figure 1B and D*). Furthermore, immunocytochemistry experiments confirmed that STX2, as well as STX1A, is strongly expressed in our model neurons (*Figure 1—figure supplement 1*).

## STX2 does not support synchronous Ca²⁺-evoked release, the RRP size, nor the spontaneous release clamp

Next, we analyzed the electrophysiological properties of autaptic neurons rescued with STX1A or STX2. First, STX2 neurons showed a dramatic reduction in the excitatory postsynaptic current (EPSC) amplitude from 4.3 nA (SEM ±0.53) to 0.7 nA (SEM ±0.16) (*Figure 1E*). We plotted the cumulative charge transfer of the EPSC over more than 550 ms after the stimulation. A rapid and synchronous release of vesicles was reflected in a much more rapid cumulative charge transfer in STX1A neurons (*Figure 1F*, black line) compared to STX2 neurons, where the cumulative charge transfer of the response was much slower (*Figure 1F*, purple line). Additionally, the half-width of the EPSC was increased more than twofold in STX2 neurons (*Figure 1G*). This resonates with previous studies that showed a decrease in the speed of secretion of STX2-expressing non-neuronal cells (*Abonyo et al., 2004*). Overall, our results indicate that STX2 does not support synchronous Ca²⁺-evoked release, but a slow asynchronous response. We also evaluated synaptic vesicle priming by applying a hypertonic sucrose solution (500 mM) for 5 s and measuring the charge of the response (*Rosenmund and Stevens, 1996*) and found that STX2 neurons exhibit a 75% reduction in the RRP of vesicles compared to STX1A neurons (*Figure 1H*). This suggests that STX2 does not support synaptic vesicle priming to the same extent as STX1A. STX2-expressing neurons also displayed more than twofold increase in the frequency of spontaneous release (mEPSC) relative to STX1A neurons (*Figure 1I*) and a 55% increase in the probability of release (PVR), calculated by dividing the EPSC charge by the RRP charge (*Figure 1J*). When we normalized the mEPSC frequency to the RRP size, we found a nearly 15-fold increase in the spontaneous vesicular release rate in STX2 compared to STX1A neurons (*Figure 1K*). Finally, no change was observed in the paired-pulse ratio (PPR) between STX1A and STX2 groups (*Figure 1L*). Notably, the difficulty of analysis given the small values of the release of the second pulse for STX2 neurons makes it hard to interpret this data. The increase in the spontaneous vesicular release rate and the PVR indicates that STX2-containing SNARE complexes render vesicles more fusogenic. Taken together, this data suggests that STX2 neurons follow a fusion energetic landscape that differs from STX1A and resembles constitutive versus regulated release (reviewed by *Sørensen, 2009*).

## The C-terminal half of the SNARE domain of STX1A has a regulatory effect on the RRP size and spontaneous release

Previous structure–function studies have shown the critical role of the distinct domains of STX1 in different aspects of synaptic vesicle fusion. However, what characterizes the protein is its SNARE domain. Not only does the SNARE domain of STX1A form one of the four helixes of the SNARE complex (*Sutton et al., 1998*), it also interacts with several regulatory proteins that modulate priming and fusion, such as SM protein Munc18-1, calcium sensor synaptotagmin-1 (SYT1), and accessory protein complexin. Additionally, the SNARE domain has been compartmentalized into two functional subdomains: N-terminus, key in priming the vesicles to the plasma membrane and C-terminus, critical

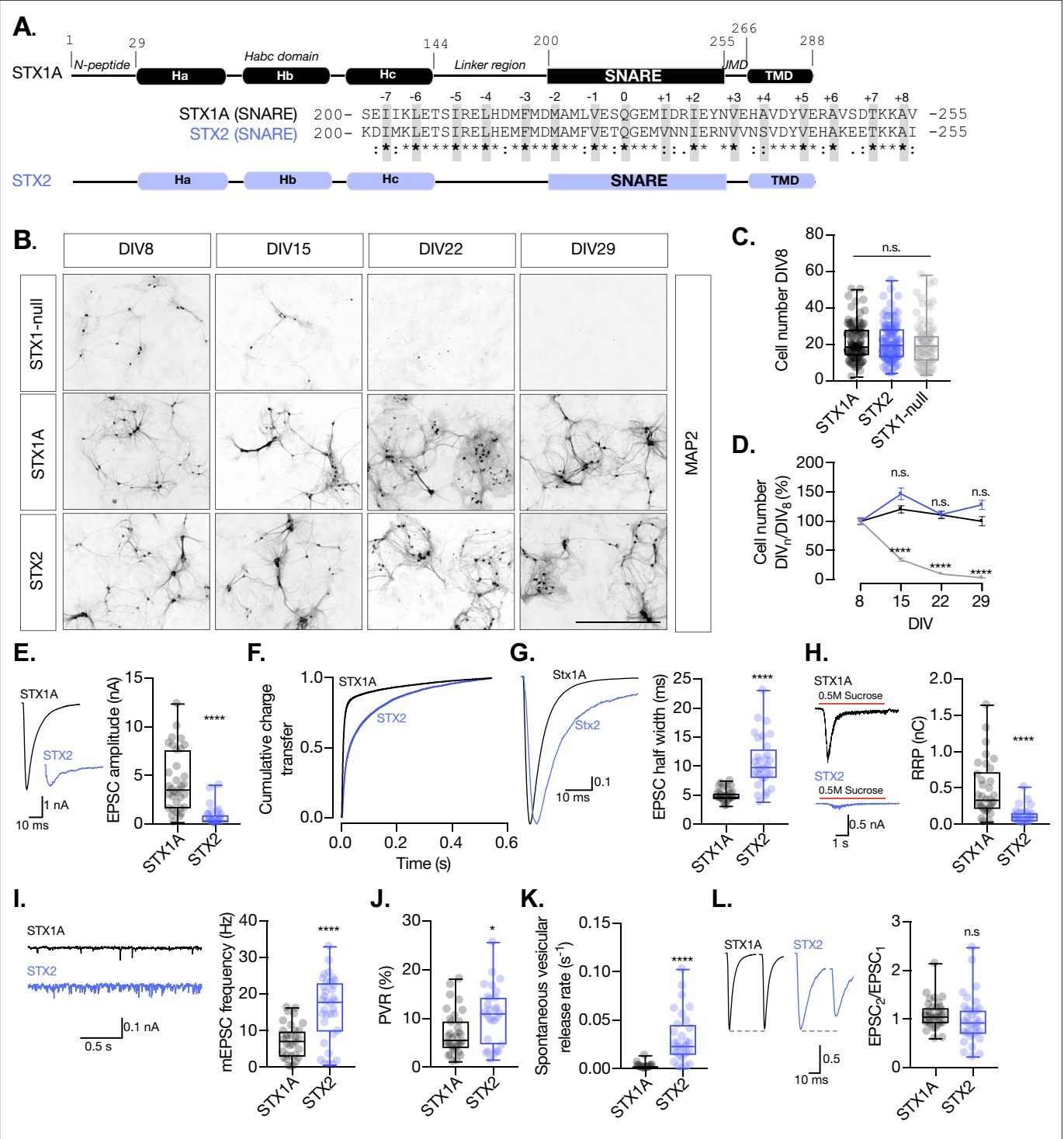

**Figure 1.** STX2 supports neuronal viability but does not rescue synchronous evoked release, the readily releasable pool (RRP), or the clamping of spontaneous release. (**A**) STX1A and STX2 domain structure scheme and SNARE domain sequence alignment (68% homology). Layers are highlighted in gray. (**B**) Example images of high-density cultured STX1-null hippocampal neurons rescued with GFP (STX1-null), STX1A, or STX2 (from top to bottom) at days in vitro (DIV)8, DIV15, DIV22, and DIV29 (from left to right). Immunofluorescent labeling of MAP2. Scale bar: 500 μm. (**C**) Quantification of total number of neurons at DIV8 of each group. (**D**) Quantification of the percentage of the surviving neurons at DIV8, DIV15, DIV22, and DIV29 normalized to the number of neurons at DIV8 in the same group. (**E**) Example traces (left) and quantification of excitatory postsynaptic current (EPSC) amplitude (right) from autaptic Stx1A-null hippocampal mouse neurons rescued with STX1A or STX2. (**F**) Quantification of the cumulative charge transfer of the EPSC from the onset of the response until 0.55 s after. (**G**) Example traces of normalized EPSC to their peak amplitude (left) and quantification of the half-width of the EPSC (right). (**H**) Example traces (left) and quantification of the response induced by a 5 s 0.5 M application of sucrose, which represents

*Figure 1 continued on next page*

*Figure 1 continued*

the RRP of vesicles. (**I**) Example traces (left) and quantification of the frequency of the miniature EPSC (mEPSC) (right). (**J**) Quantification of the vesicle release probability (PVR) as the ratio of the EPSC charge over the RRP charge (PVR). (**K**) Quantification of the spontaneous vesicular release rate as the ratio between the mEPSC frequency and number of vesicles in the RRP. (**L**) Example traces (left) and the quantification of a 40 Hz paired-pulse ratio (PPR). In (**D**), data points represent mean ± SEM. In (**D, E–L**), data is shown as a whisker-box plot. Each data point represents single observations, middle line represents the median, boxes represent the distribution of the data, and external data points represent outliers. In (**C, D**), significances and p-values of data were determined by nonparametric Kruskal–Wallis test followed by Dunn's post hoc test; *p≤0.05, **p≤0.001, ***p≤0.001, ****p≤0.0001. In (**E–L**), significances and p-values of data were determined by nonparametric Mann–Whitney test and unpaired two-tailed *t*-test; *p≤0.05, **p≤0.01, ***p≤0.001, ****p≤0.0001. All data values are summarized in *Figure 1—source data 1*.

The online version of this article includes the following source data and figure supplement(s) for figure 1:

**Source data 1.** Quantification of the neuronal density at different time intervals and quantification of neurotransmitter release parameters of STX1-null neurons transduced with STX1A and STX2.

**Figure supplement 1.** STX2 is expressed in STX1-null hippocampal neurons.

**Figure supplement 1—source data 1.** Quantification of STX1A and STX2 levels in STX1-null neurons.

**Figure supplement 1—source data 2.** Whole SDS-PAGE image represented in *Figure 1—figure supplement 1*.

---

for the fusion process (*Gao et al., 2012*; *Schotten et al., 2015*; *Sørensen et al., 2006*; *Walter et al., 2010*; *Weber et al., 2010*). Our neuronal survival test shows that we can utilize STX2 for electrophysiological studies. To confine our studies to the SNARE domain, we used a chimeric approach, where we exchanged either the entire SNARE domain or the N- or C-terminal halves of the SNARE domain (N-terminal from E/D200-Q226; C-terminal from Q226-V/I255) between STX1A and STX2 (*Figure 2A*). Whereas the N-terminal halves of STX1A and STX2 share an 81% homology, their C-terminal halves only share 60% homology (*Figure 2A*), and each half potentially contributes to a different step in vesicle fusion (*Gao et al., 2012*; *Sørensen et al., 2006*; *Weber et al., 2010*).

Replacement of the full-SNARE domain (STX1A-2(SNARE)) or the C-terminal half (STX1A-2(Cter)) of the SNARE domain of STX1A with the same domain from STX2 resulted in a reduction in the EPSC amplitude (*Figure 2B*). However, there was no change in the cumulative charge transfer (*Figure 2C*) or the kinetic parameters of the EPCS, such as half-width (*Figure 2D*) or rise and decay time (*Figure 2—figure supplement 1A and B*), in any of the groups. This suggests that the SNARE domain of STX1A may not be directly involved in the regulation of the kinetics of the release. Our analysis also showed that neurons expressing STX1A-2(SNARE) or STX1A-2(Cter) exhibited a nearly threefold increase in the mEPSC frequency (*Figure 2E*), potentially indicating an unclamping of spontaneous release. Additionally, the total charge transfer in response to a hypertonic sucrose application was significantly decreased in these two groups from 0.5 nC (SEM ±0.06) to 0.18 nC (SEM ±0.02) and 0.2 nC (SEM ±0.02), respectively, compared to STX1A WT, indicating fewer primed synaptic vesicles and a smaller RRP (*Figure 2F*). Notably, the common modification of these STX1A chimera groups is the addition of the C-terminal half of the SNARE domain of STX2. These results suggest that the C-terminal half of the SNARE domain of STX1A may be directly involved in the clamping mechanism for the spontaneous release of synaptic vesicles and may also play a role in the modulation of the RRP. However, it should be noted that we also observed a significant increase from 6.84 Hz (SEM ±0.62) to 11.8 Hz (SEM ±1.28) in the mEPSC frequency of STX1A-2(Nter)-expressing neurons (*Figure 2E*). Therefore, we cannot rule out the possibility that the N-terminal half of the SNARE domain may also play a role in the regulation of spontaneous release.

We next examined release efficacy in the STX1A-STX2 SNARE domain chimeras. STX1A-2(SNARE) showed a 39% increase and STX1A-2(Cter) showed a 42% increase in the PVR (*Figure 2G*), which indicates that the reduction in the EPSC is not proportional to the decrease in the RRP. This suggests that the changes we observed could be due to a faulty priming mechanism, but it could also mean that for these two groups there is a change in an additional mechanism that goes beyond the RRP size regulation and increases the efficacy of the Ca²⁺-evoked release. Finally, STX1A-2(SNARE) and STX1A-2(Cter) had a 17- and 8-fold increase, respectively, in the spontaneous vesicular release rate (*Figure 2H*), a 40% decrease in the PPR (*Figure 2I*), and increased depression in a 10 Hz train stimulation (*Figure 2J*). Taken together our results suggest that the C-terminal half of the SNARE domain of STX1A is involved in the regulation of the efficacy of Ca²⁺-evoked release, the formation of the RRP, and the clamping of spontaneous release.

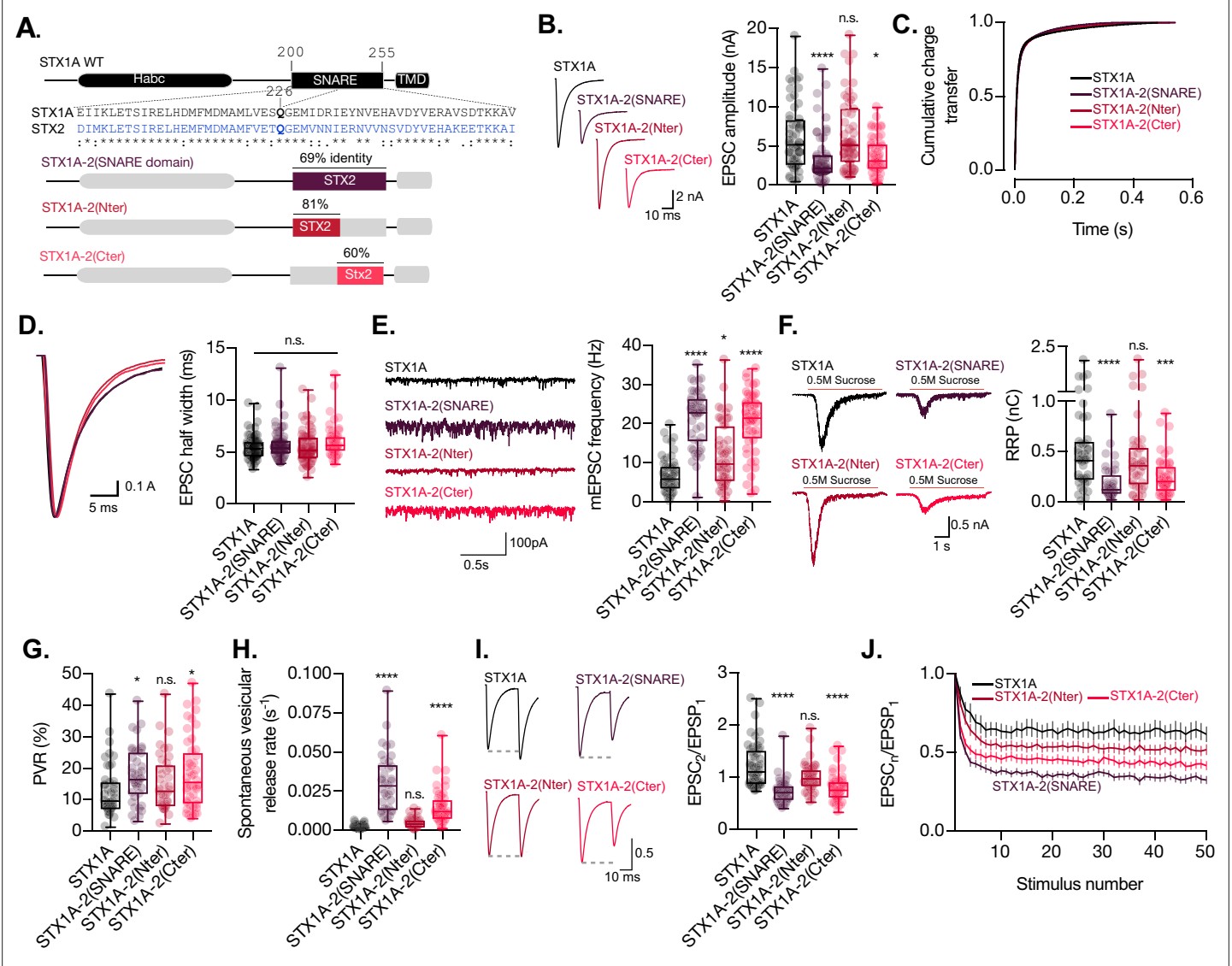

**Figure 2.** The C-terminal half of the SNARE domain of STX1A has a regulatory effect on the readily releasable pool (RRP), spontaneous release, and both, and N- and C-terminus have a role in the regulation of efficacy of Ca²⁺-evoked release. (**A**) STX1A WT and chimera domain structure scheme, sequence alignment of STX1A and STX2 SNARE domain, and percentage homology between both SNARE domains. (**B**) Example traces (left) and quantification of the excitatory postsynaptic current (EPSC) amplitude (right) from autaptic STX1A-null hippocampal mouse neurons rescued with STX1A, STX1A-2(SNARE), STX1A-2(Nter), or STX1A-2(Cter). (**B**) Quantification of the cumulative charge transfer of the EPSC from the onset of the response until 0.55 s after. (**C**) Example traces of normalized EPSC to their peak amplitude (left) and quantification of the half-width of the EPSC (right). (**D**) Example traces (left) and quantification of the frequency of the miniature EPSCs (mEPSC) (right). (**E**) Example traces (left) and quantification of the response induced by a 5 s 0.5 M application of sucrose, which represents the RRP of vesicles. (**F**) Quantification of the vesicle release probability (PVR) as the ratio of the EPSC charge over the RRP charge (PVR). (**G**) Quantification of the spontaneous vesicular release rate as the ratio between the mEPSC frequency and number of vesicles in the RRP. (**H**) Example traces (left) and the quantification of a 40 Hz paired-pulse ratio (PPR). (**I**) Quantification of STP measured by 50 stimulations at 10 Hz. In (**B, D–I**), data is shown as a whisker-box plot. Each data point represents single observations, middle line represents the median, boxes represent the distribution of the data, where the majority of the data points lie, and external data points represent outliers. In (**J**), data represents the mean ± SEM. Significances and p-values of data were determined by nonparametric Kruskal–Wallis test followed by Dunn's post hoc test; *p≤0.05, **p≤0.01, ***p≤0.001, ****p≤0.0001. All data values are summarized in **Figure 2—source data 1**.

The online version of this article includes the following source data and figure supplement(s) for figure 2:

**Source data 1.** Quantification of neurotransmitter release parameters of STX1-null neurons transduced with STX1A, STX1A-2(SNARE), STX1A-2(Nter) and STX1A-2(Cter).

**Figure supplement 1.** Quantification of kinetic parameters of the excitatory postsynaptic current (EPSC) and the miniature EPSC (mEPSC) of autaptic STX1A-null hippocampal mouse neurons rescued with STX1A, STX1A-2(SNARE), STX1A-2(Nter), or STX1A-2(Cter).

*Figure 2 continued on next page*

*Figure 2 continued*

**Figure supplement 1—source data 1.** Quantification of neurotransmitter release parameters of STX1-null neurons transduced with STX1A, STX1A-2(SNARE), STX1A-2(Nter) and STX1A-2(Cter).

## The insertion of the full-length SNARE domain or the C-terminal half of the SNARE domain of STX1A into the STX2 backbone has an impact on the kinetics of evoked release, RRP size, and spontaneous release

In the same manner as before, we wanted to test whether any of the electrophysiological properties that differ between the two syntaxin isoforms are attributable to the SNARE domain itself and can be transferred to STX2. For this, we generated chimeric proteins by introducing the entire SNARE domain of STX1A (STX2-1A(SNARE)), its N- terminal half (STX2-1A(Nter)), or its C-terminal half (STX2-1A(Cter)) into STX2 (*Figure 3A*). Not only does this chimeric approach allow us to isolate certain functions of the SNARE domain, but it can also help us determine whether the function of the SNARE domain of STX1A depends on domains or sequences present in its natural structure.

When recording from the STX2-STX1A SNARE domain chimera-expressing neurons, we observed that we were able to partially rescue the EPSC amplitude when STX2 contained the entire SNARE domain of STX1A (STX2-1A(SNARE)) compared to STX2, from 0.35 nA (SEM ±0.044) to 2.6 nA (SEM ±0.28), respectively, although not to STX1A WT levels 4.06 nA (SEM ±0.31). Additionally, STX2-1A(Cter) showed a slight, albeit significant, rescue of the amplitude of the $Ca^{2+}$-evoked response to 1.13 nA (SEM ±0.28). STX2-1A(Nter) also showed a 50% increase in the $Ca^{2+}$-evoked release to 0.68nA (SEM ±0.08) (*Figure 3B*), suggesting an insufficient rescue of the evoked response. To analyze the kinetics of release in STX2-chimeras, we plotted the cumulative charge transfer of the EPSC over time for each group and quantified kinetic parameters such as half-width, rise time, and decay time. STX2 and STX2-1A(Nter) exhibited the slowest cumulative charge transfer (*Figure 3C*, blue and light blue lines, respectively) as well as the highest half-width (*Figure 3D*) and rise and decay time (*Figure 3—figure supplement 1A and B*). These findings indicate that the N-terminal half of the SNARE domain of STX1A is not enough to support the fast kinetics of the response. However, the introduction of the entire SNARE domain or the C-terminal half rescued the speed and synchronicity of the EPSC to almost STX1A WT levels (*Figure 3C*, dark and light turquoise lines, respectively, *Figure 3D*, and *Figure 3—figure supplement 1A and B*). Notably, the speed of release did not change in any STX1A-chimera (*Figure 2C and D*), suggesting that there is a dominant regulatory domain which supports the fast kinetics of the response outside of the SNARE domain in STX1A, such as the JMD or TMD (*Vardar et al., 2022*). However, the C-terminal half of STX1A alone is sufficient to evoke faster responses in STX2. These results support major regulatory differences of the domains outside of the SNARE domain between the isoforms, which should not be discarded in the interpretation of our results.

As reported previously, our STX1A chimeric analysis suggested a clamping function of the C-terminal half of the SNARE domain of STX1A (*Figure 2E*). In line with these findings, STX2-1A(Cter) showed a significant reduction in the mEPSC frequency to almost STX1A WT levels. STX2-1A(SNARE) unexpectedly showed a 47% increase in the spontaneous release frequency compared to STX2, which was also observed in STX2-1A(Nter) (*Figure 3E*). Additionally, we found that STX2-1A(SNARE) and STX2-1A(Cter) could rescue the RRP to around double of what we measured from STX2 and STX2-1A(Nter) (*Figure 3F*), albeit not significant. This trend supports our hypothesis that the C-terminal half of the SNARE domain of STX1A potentially plays a role in the regulation in vesicle priming, also revealed by STX1A-chimeras (*Figure 2F*). Furthermore, STX2-1A(SNARE) and STX2-1A(Nter) had an increased PVR (*Figure 3G*), no change in the release rate (*Figure 3H*), and an increase in short-term depression during 10 Hz train stimulation (*Figure 3J*), while only STX2-1A(SNARE) in the PPR (*Figure 3I*), compared to STX2 WT. On the other hand, STX2-1A(Cter) showed no change in the PVR compared to STX2 or STX1A WT, a 4.5-fold decrease in the spontaneous release rate (STX1A WT has a 12-fold decrease compared to STX2), and an increase in the short-term depression in a 10 Hz train stimulation, albeit the difficulty of analysis of this data given the small values of the release (*Figure 3J*). These data indicate that the suppression of spontaneous release and the regulation of the RRP rely on the C-terminal half of the SNARE domain of STX1A, while the efficacy of $Ca^{2+}$-evoked release depends on the integrity of the entire SNARE domain (*Figure 3G, I, and J*).

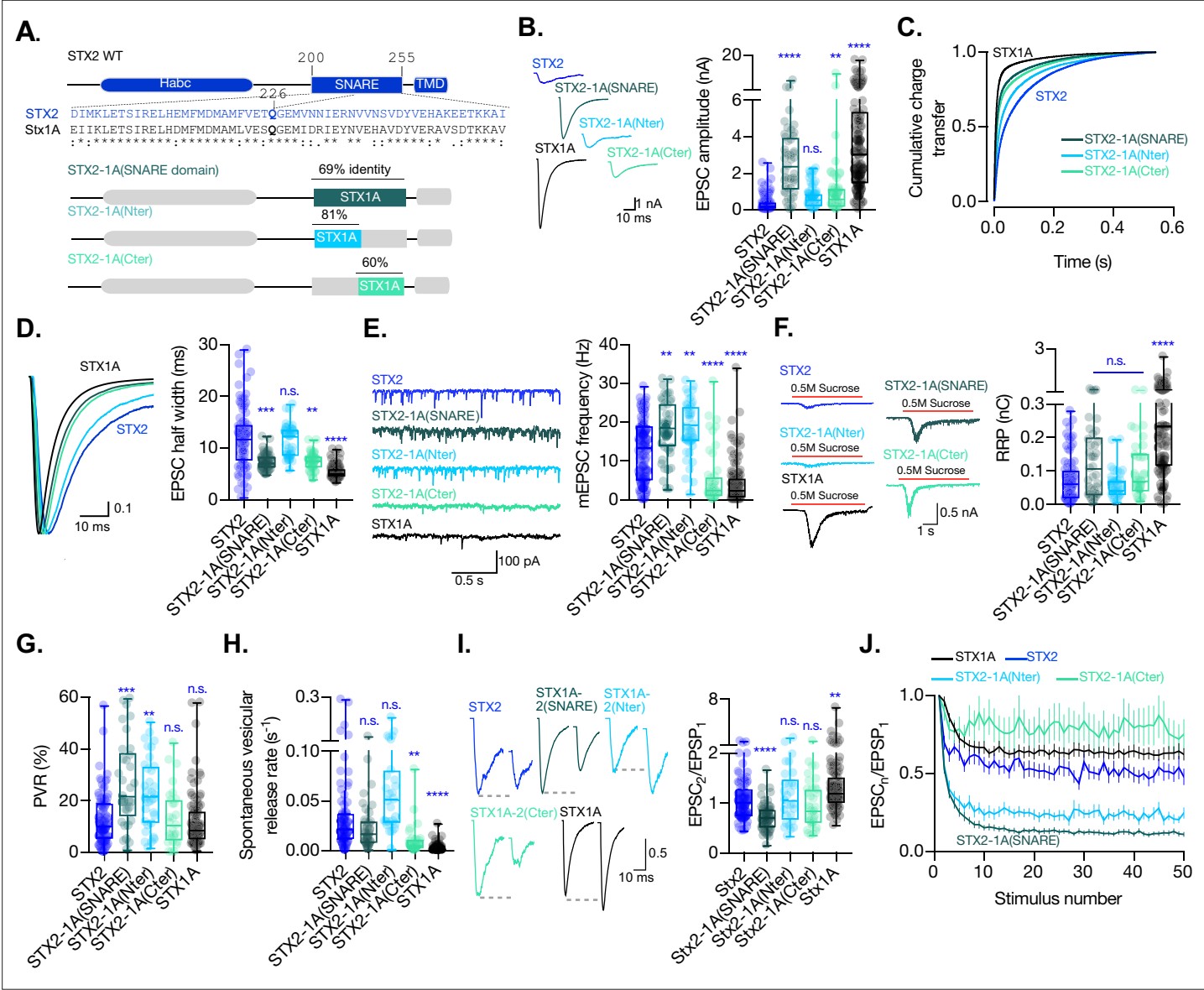

**Figure 3.** The C-terminal half of the SNARE domain of STX1A has a regulatory effect on the spontaneous release and the readily releasable pool (RRP), and the speed of Ca²⁺-evoked release depends on the integrity of the SNARE domain. (**A**) STX2 WT and chimera domain structure scheme, sequence alignment of STX2 and STX1A SNARE domain, and percentage homology between both SNARE domains. (**B**) Example traces (left) and quantification of the excitatory postsynaptic current (EPSC) amplitude (right) from autaptic STX1A-null hippocampal mouse neurons rescued with STX2, STX2-1A(SNARE), STX2-1A(Nter), or STX2-1A(Cter). (**C**) Quantification of the cumulative charge transfer of the EPSC from the onset of the response until 0.55 s after. (**D**) Example traces of normalized EPSC to their peak amplitude (left) and quantification of the half-width of the EPSC (right). (**E**) Example traces (left) and quantification of the frequency of the miniature EPSC (mEPSC) (right). (**F**) Example traces (left) and quantification of the response induced by a 5 s 0.5 mM application of sucrose, which represents the RRP of vesicles. (**G**) Quantification of the vesicle release probability (PVR) as the ratio of the EPSC charge over the RRP charge (PVR). (**H**) Quantification of the spontaneous vesicular release rate as the ratio between the mEPSC frequency and number of vesicles in the RRP. (**I**) Example traces (left) and the quantification of a 40 Hz paired-pulse ratio (PPR). (**J**) Quantification of STP measured by 50 stimulations at 10 Hz. In (**B, D-I**), data is shown as whisker-box plot. Each data point represents single observations, middle line represents the median, boxes represent the distribution of the data, where the majority of the data points lie, and external data points represent outliers. Significances and p-values of data were determined by nonparametric Kruskal–Wallis test followed by Dunn's post hoc test; *p≤0.05, **p≤0.01, ***p≤0.001, ****p≤0.0001. All data values are summarized in *Figure 3—source data 1*.

The online version of this article includes the following source data and figure supplement(s) for figure 3:

**Source data 1.** Quantification of neurotransmitter release parameters of STX1-null neurons transduced with STX2, STX2-1A(SNARE), STX2-1A(Nter) and STX2-1A(Cter).

*Figure 3 continued on next page*

Figure 3 continued

**Figure supplement 1.** Quantification of kinetic parameters of the excitatory postsynaptic current (EPSC) and the miniature EPSC (mEPSC) of autaptic STX1A-null hippocampal mouse neurons rescued with STX2, STX2-1A(SNARE), STX2-1A(Nter), or STX2-1A(Cter).

**Figure supplement 1—source data 1.** Quantification of neurotransmitter release parameters of STX1-null neurons transduced with STX2, STX2-1A(SNARE), STX2-1A(Nter) and STX2-1A(Cter).

## The insertion of the full-length SNARE domain or the C-terminal half of the SNARE domain of STX1A into the Stx2 backbone rescues Munc18-1 levels at the synapse

Munc18-1 binds not only to the N-terminal regions of STX1A (*Burkhardt et al., 2008*; *Khvotchev et al., 2007*; *Meijer et al., 2018*) but also to the SNARE complex (*Baker et al., 2015*; *Jiao et al., 2018*; *Stepien et al., 2022*). The expression levels of Munc18-1 and STX1A influence each other, and their interaction is important for priming and release (*Gerber et al., 2008*; *Vardar et al., 2016*). Thus, we analyzed the exogenous expression of STX1A, STX1A-2(SNARE), STX1A-2(Nter), and STX1A-2(Cter) and the endogenous expression of Munc18-1 at the synapse (*Figure 4*). We used VGlut1 as the synaptic marker in the co-localization assays. We found that the introduction of the SNARE domain of STX2 into STX1A (STX1A-2(SNARE)) increased the expression of this construct compared to STX1A WT while the rest showed similar levels (*Figure 4B*). Additionally, the exchange of the entire SNARE domain or the C-terminus of STX1A with that of STX2 caused an increase in the levels of endogenous Munc18-1 at the synapse, which may be relative to the increase in STX1A-construct levels (*Figure 4C*). Because the loss or overexpression of Munc18-1 affects the properties of secretion and release (*Oh et al., 2012*; *Toonen et al., 2006*; *Verhage et al., 2000*; *Voets et al., 2001*), we wanted to determine whether changes in the levels of Munc18-1 may correlate with the changes in neurotransmitter release from the neurons which carry the exogenous STX1A-chimeric constructs. We observed that higher levels of Munc18-1 seem to correlate with less Ca$^{2+}$-evoked release (*Figure 4E*), and a smaller RRP (*Figure 4F*) but higher release efficacy (*Figure 4G*) and release rates (*Figure 4H*), which suggests it may be the cause of these changes in the release parameters.

Additionally, we quantified the levels of exogenous expression of STX2, STX2-1A(SNARE), STX2-1A(Nter), and STX2-1A(Cter) and the corresponding levels of endogenous Munc18-1 levels at the synapse (*Figure 5*). We found that all STX2 constructs were expressed at similar levels at the synapse (co-localized to VGlut1 puncta) (*Figure 5A and B*). However, STX2-1A(SNARE) and STX2-1A(Cter) showed a trend toward increase in their expression (*Figure 5B and C*). We then quantified the endogenous levels of Munc18-1 at the synapse in these neurons and found a decrease in Munc18-1 levels when expressing STX2 WT compared to STX1A WT. Interestingly, Munc18-1 levels were rescued when either the entire SNARE domain (STX2-1A(SNARE)) or the C-terminal half (STX2-1A(Cter)) of STX1A were present (*Figure 5E and F*) compared to STX1A WT neurons. The interaction between Munc18-1 and the SNARE domain, which is thought to be important for templating SNARE complex assembly (*Baker et al., 2015*; *Stepien et al., 2022*), may be directly involved in rescuing some of the electrophysiological parameters observed in STX2 chimeras. To investigate this, we correlated the levels of Munc18-1 with electrophysiological phenotypes observed in neurons expressing the STX2-chimeras (*Figure 5G–J*). Our superposition of phenotypes revealed that higher levels of Munc18-1, observed in STX1A WT and in the STX2-1A(Cter) neurons, showed a trend in increasing Ca$^{2+}$-evoked release (*Figure 5G*) and the RRP (*Figure 5H*) while the effects on PVR were more randomly distributed (*Figure 5I*). Additionally, we observed that higher release rates correspond to groups, which show less amounts of Munc18-1 at the synapse (*Figure 5J*). Taking together these results it seems like we have contradictory observations: while increased levels of Munc18-1 in the STX1A background seem to correlate with parameters that depict an increase in the fusogenicity of the vesicles (*Figure 4*), an increase in the levels of Munc18-1 in the STX2 background correlates to release properties that incline toward a decreased fusogenicity. In summary, although Munc18-1 levels do depend on the construct used and may be causative of changes in certain electrophysiological characteristics, we may not be able to explain these phenotypes by the levels of Munc18-1.

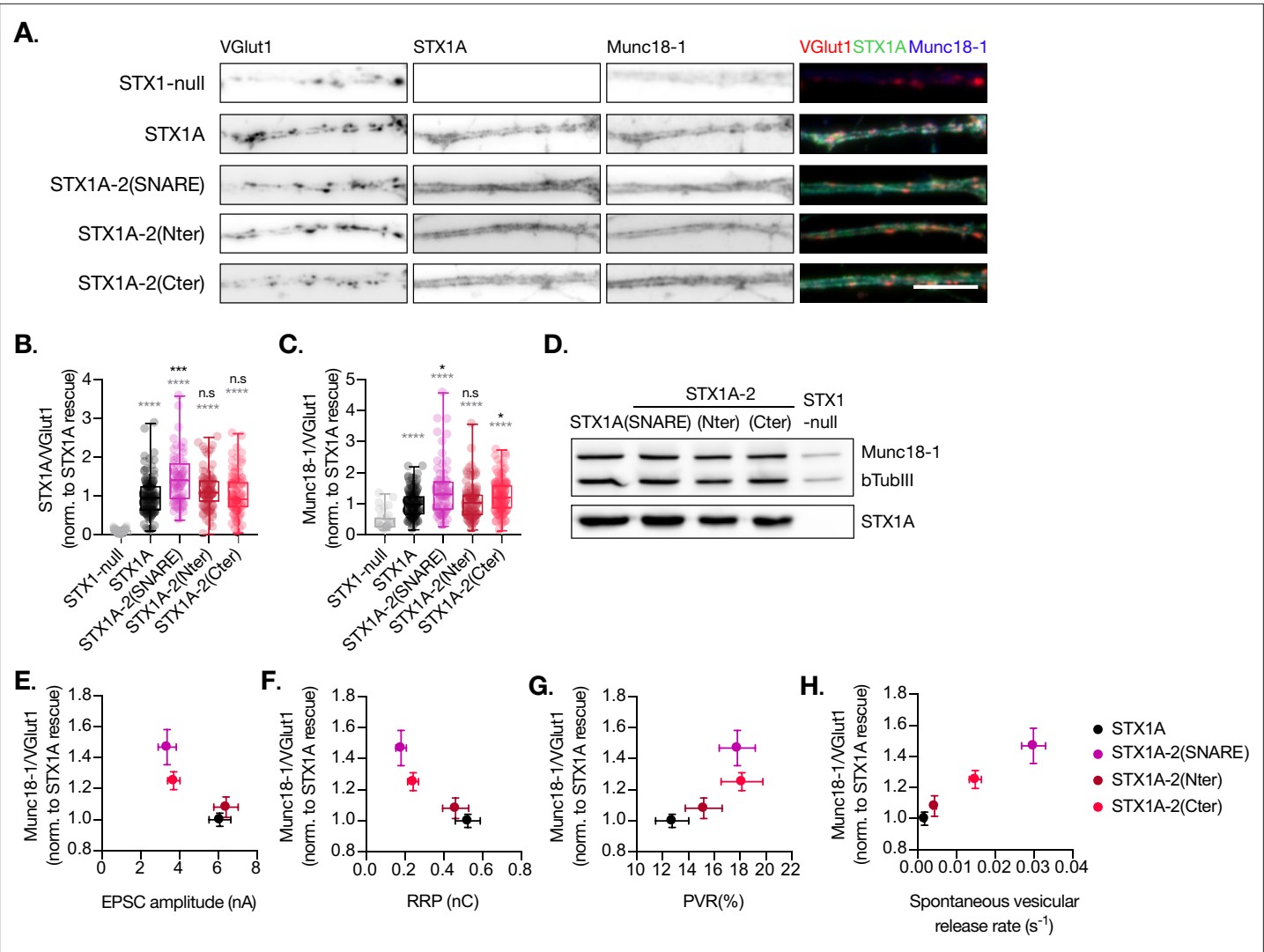

**Figure 4.** Quantification of STX1A and Munc18-1 levels at the synapse. (**A**) Example images of Stx1-null neurons plated in high-density cultures and rescued with STX1A, STX1A-2(SNARE), STX1A-2(Nter), or STX1A-2(Cter) or GFP (STX1-null) as negative control. Neurons were fixed between days in vitro (DIV)14–16. Cultures were stained with fluorophore-labeled antibodies that recognize VGlut1 (red in merge), STX1A (green in merge), and Munc18-1 (blue in merge), from left to right. Scale bar: 10 μm. (**B**) Quantification of the immunofluorescent intensity of STX1A normalized to the intensity of the same VGlut1-labeled ROIs. Values were normalized to values in the STX1A rescue group. (**C**) Quantification of the immunofluorescent intensity of Munc18-1 normalized to the intensity of the same VGlut1-labeled ROIs. Values were normalized to values in the STX1 rescue group. (**D**) SDS-PAGE of the electrophoretic analysis of neuronal lysates obtained from each experimental group. Proteins were detected using antibodies that recognize β-tubuline III as loading control, STX1A, and Munc18-1. (**E**) Correlation between Munc18-1 values and excitatory postsynaptic current (EPSC) amplitude of Stx1-null neurons expressing STX1A, STX2, STX2-1A(SNARE), STX2-1A(Nter), or STX2-1A(Cter). (**A**) Correlation between Munc18-1 values and readily releasable pool (RRP). (**B**) Correlation between Munc18-1 values and probability of release (PVR). (**C**) Correlation between Munc18-1 values and spontaneous vesicular release rate. In (**B, C**), data is shown as a whisker-box plot. Each data point represents single ROIs, middle line represents the median, boxes represent the distribution of the data, where the majority of the data points lie, and external data points represent outliers. In (**E–H**), each data point is the correlation of the mean ± SEM. Significances and p-values of data were determined by nonparametric Kruskal–Wallis test followed by Dunn's post hoc test; *p≤0.05, **p≤0.01, ***p≤0.001, ****p≤0.0001. All data values are summarized in *Figure 4—source data 1*.

The online version of this article includes the following source data for figure 4:

**Source data 1.** Quantification of STX1A and Munc18 levels in STX1-null neurons transduced with STX1A, STX1A-2(SNARE), STX1A-2(Nter) and STX1A-2(Cter).

**Source data 2.** Whole SDS-PAGE image represented in *Figure 4*.

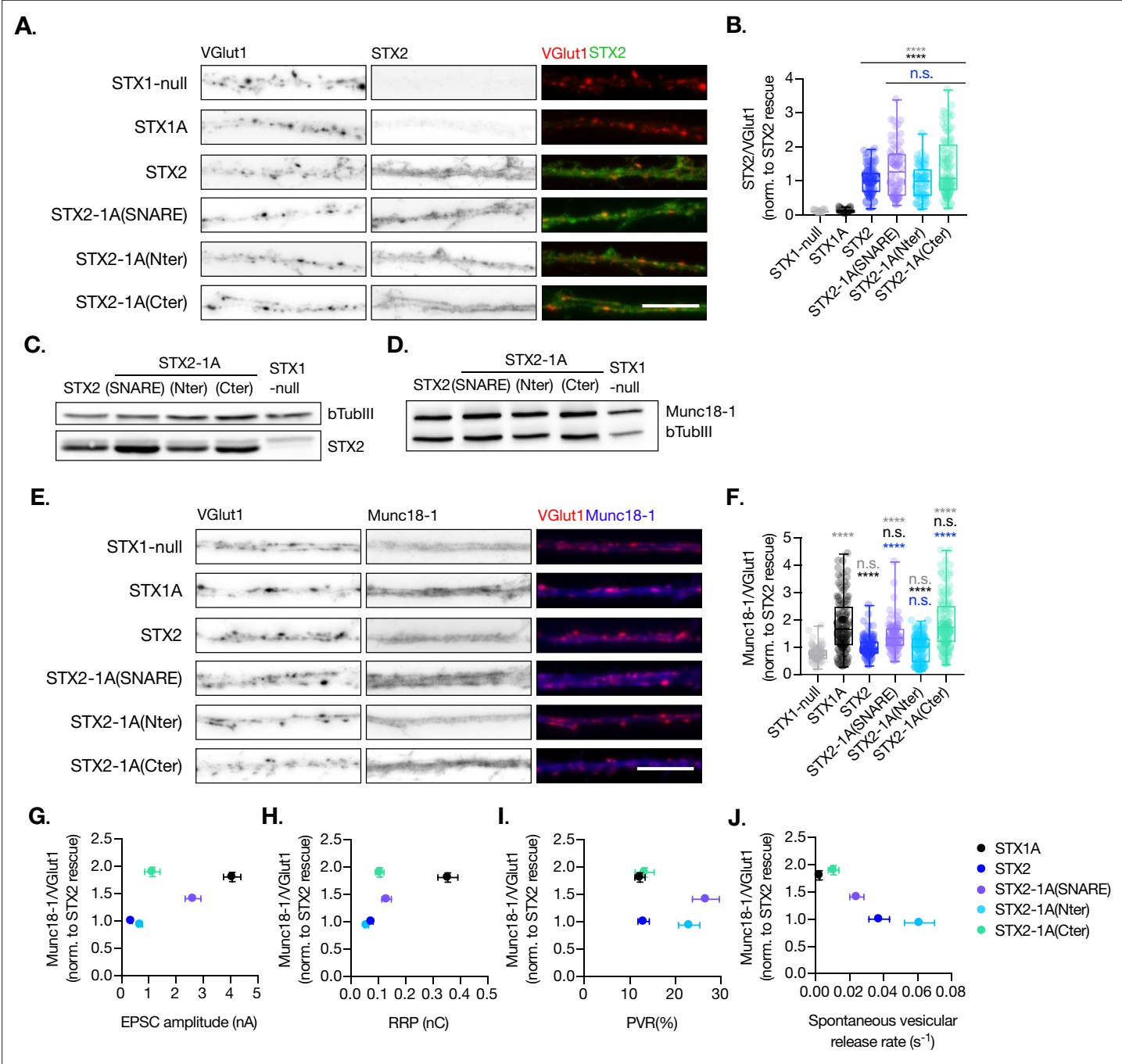

**Figure 5.** Quantification of STX2 and Munc18-1 levels at the synapse. (**A**) Example images of Stx1-null neurons plated in high-density cultures and rescued with STX1A, STX2, STX2-1A(SNARE), STX2-1A(Nter), STX2-1A(Cter), or GFP (STX1-null) as negative control. Neurons were fixed between days in vitro (DIV14–16 . (**A**) Example images of neurons stained with fluorophore-labeled antibodies that recognize VGlut1 (red in merge) and STX2 (green in merge), from left to right. Scale bar: 10 µm. (**B**) Quantification of the immunofluorescent intensity of STX2 normalized to the intensity of the same VGlut1-labeled ROIs. Values were normalized to values in the STX2 rescue group. (**C**) SDS-PAGE of the electrophoretic analysis of neuronal lysates obtained from each experimental group. Proteins were detected using antibodies that recognize β-tubuline III as loading control and STX2. (**D**) SDS-PAGE of the electrophoretic analysis of neuronal lysates obtained from each experimental group. Proteins were detected using antibodies that recognize β-tubuline III and Munc18-1. (**E**) Example images of neurons stained with fluorophore-labeled antibodies that recognize VGlut1 (red in merge) and Munc18-1 (blue in merge), from left to right. Scale bar: 10 µm. (**F**) Quantification of the immunofluorescent intensity of Munc18-1 normalized to the intensity of the same VGlut1-labeled ROIs. Values were normalized to values in the STX2 rescue group. (**G**) Correlation between Munc18-1 values and excitatory postsynaptic current (EPSC) amplitude of STX1-null neurons expressing STX1A, STX2, STX2-1A(SNARE), STX2-1A(Nter), or STX2-1A(Cter). (**H**) Correlation between Munc18-1 values and readily releasable pool (RRP). (**I**) Correlation between Munc18-1 values and probability of

*Figure 5 continued on next page*

Figure 5 continued

release (PVR). (**J**) Correlation between Munc18-1 values and spontaneous vesicular release rate. In (**B, F**), data is shown as a whisker-box plot. Each data point represents single ROIs, middle line represents the median, boxes represent the distribution of the data, where the majority of the data points lie, and external data points represent outliers. In (**G–J**), each data point is the correlation of the mean ± SEM. Significances and p-values of data were determined by nonparametric Kruskal–Wallis test followed by Dunn's post hoc test; *p≤0.05, **p≤0.01, ***p≤0.001, ****p≤0.0001. All data values are summarized in *Figure 5—source data 1*.

The online version of this article includes the following source data for figure 5:

**Source data 1.** Quantification of STX2 and Munc18 levels in STX1-null neurons transduced with STX2, STX2-1A(SNARE), STX2-1A(Nter) and STX2-1A(Cter).

**Source data 2.** Whole SDS-PAGE image represented in *Figure 5*.

## The residues on the outer surface of the C-terminal half of the SNARE domain are crucial in the regulation of spontaneous release and the RRP

So far, we have highlighted the importance of the C-terminal half of the SNARE domain of STX1A in the regulation of the RRP size and for the clamping of spontaneous release. Our previous study that aligned STX1A and different STX3A and STX3B chimeric constructs also suggested the involvement of the C-terminus half in the spontaneous release clamping mechanisms, but only through elimination (*Vardar et al., 2022*). These data compelled us to examine whether spontaneous release regulatory mechanisms can be attributed to individual or sequentially paired amino acid residues in the C-terminal half of the SNARE domain of STX1A. To test this, we introduced single- or double-point mutations in the C-terminus of the SNARE domain of STX1A, targeting residues that differ from STX2 in their charge or polarity and that are located on the outer surface of the SNARE complex. It is important to note that electrostatic charges play a critical role in the interaction of STX1A's SNARE domain with other proteins and the membrane (*Ruiter et al., 2019*). Thus, we hypothesized that if the charge of the amino acid was not significantly altered, it may not play a significant role in the observed phenotypic differences between STX1A and STX2. Given this, we generated single-point mutations (STX1A$^{D231N}$, STX1A$^{R232N}$, STX1A$^{Y235R}$, STX1A$^{E238V}$, STX1A$^{V248K}$, and STX1A$^{S249E}$) and double-point mutations (STX1A$^{D231N,R232N}$ and STX1A$^{V248K,S249E}$) if the residues were in sequential positions and analyzed their expression at the synapse and the corresponding levels of Munc18-1 at the synapse (*Figure 6A*, *Figure 6—figure supplement 1*). Notably, we excluded STX1A$^{A240S}$, which is present in STX1B, redundant in function to STX1A (*Vardar et al., 2016*) and STX1A$^{R246}$ and STX1A$^{D250}$, which are electrochemically similar to STX2$^{H246}$ and STX2$^{E250}$ (*Figure 6A*). For unknown reasons, the levels of some of the mutant constructs of STX1A seem to be reduced by more than 50%, which has been shown to be detrimental to release (*Arancillo et al., 2013*; *Vardar et al., 2016*), while all electrophysiological parameters showed comparable levels to STX1A WT or even an enhanced spontaneous release (*Figure 6*). For this reason, we did not explore the expression results any further. Additionally, levels in Munc18-1 of the corresponding mutants were also decreased (*Figure 6—figure supplement 1*).

The electrophysiological analysis of our point mutations showed that none of the mutations caused a change in the EPSC amplitude (*Figure 6B*). However, the double-point mutant STX1A$^{D231N,R232N}$ was the only mutant that showed a decrease in the RRP size from 0.469 nC (SEM ±0.08) to 0.19 nC (SEM ±0.03) (*Figure 6C*), as well as a twofold increase in the PVR (*Figure 6D*). This is consistent with the changes in our previous findings from the STX1A chimeric analysis, which showed a decreased RRP and an increased PVR (*Figure 2G*). All mutants showed an increase in the mEPSC frequency; however, it was not significant for STX1A$^{E238V}$ and STX1A$^{V248K}$ (*Figure 6E*). This suggests that any small alteration in the outer surface of the C-terminal half of the SNARE domain of STX1A could destabilize the clamp for spontaneous release. Therefore, the whole domain is crucial in the regulation of the synaptic vesicle clamp. The mutants with increased spontaneous release frequency also exhibited an increase in the release rate (*Figure 6F and G*), particularly those with a simultaneous increase in the mEPSC frequency and a trend toward RRP reduction. This included all the double-point mutants STX1A$^{D231N,R232N}$, STX1A$^{D231D,E238V}$, and STX1A$^{V248K,S249E}$ and the single-point mutants STX1A$^{D231}$ and STX1A$^{S249E}$. These results suggest that the more alterations the C-terminal half of the SNARE domain undergoes, the less effective it is in maintaining the stability of the vesicles in the RRP and clamping spontaneous release.

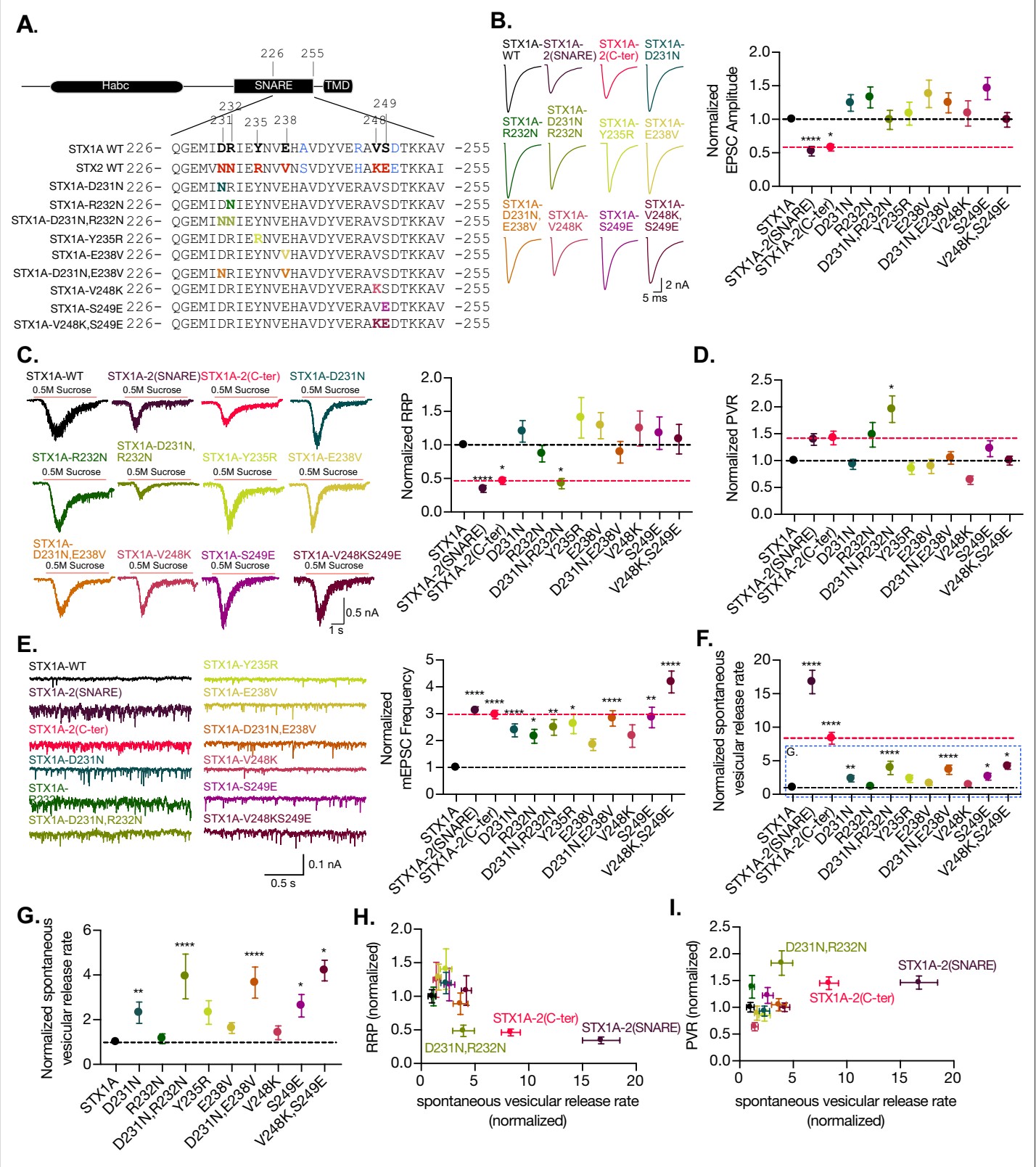

**Figure 6.** The charge in the outer-surface residues in the C-terminal half of the SNARE domain is important for clamping spontaneous release, and D231,R232 are important in the stabilization of the pool and the efficiency of Ca²⁺-evoked release. (**A**) Sequence of the C-terminal half of STX1A and STX2 and single- and double-point mutations in the sequence of STX1A WT. (**B**) Example traces (left) and quantification of the excitatory postsynaptic current (EPSC) amplitude (right) from autaptic STX1A-null hippocampal mouse neurons rescued with STX1A, STX1A^D231N, STX1A^R232N, STX1A^Y235R,

*Figure 6 continued on next page*

*Figure 6 continued*

STX1A^E238V, STX1A^V248K, STX1A^S249E STX1A^D231N,R232N, and STX1A^V248K,S249E. (**C**) Example traces (left) and quantification of the response induced by a 5 s 0.5 M application of sucrose, which represents the readily releasable pool of vesicles (RRP). (**D**) Quantification of the vesicle release probability (PVR) as the ratio of the EPSC charge over the RRP charge (PVR). (**E**) Example traces (left) and quantification of the frequency of the miniature EPSC (mEPSC) (right). (**F**) Quantification of the spontaneous vesicular release rate as the ratio between the mEPSC frequency and number of vesicles in the RRP. (**G**) Quantification of the spontaneous vesicular release rate as the ratio between the mEPSC frequency and number of vesicles in the RRP but STX1A-2(SNARE) and STX1A-2(Cter) chimeras were removed. (**H**) Correlation between the RRP and the spontaneous vesicular release rate. Both values are normalized to STX1A WT. (**I**) Correlation between the PVR and the spontaneous vesicular release rate. Values are normalized to STX1A WT. (**B–H**) All electrophysiological recording were done on autaptic neurons. Between 30 and 35 neurons per group from three independent cultures were recorded. Values from the STX1A-2(SNARE) and STX1A-2(Cter) groups were taken from the experiments done in *Figure 2* and normalized to their own STX1A WT control and used here for visual comparison. Data points represent the mean ± SEM. Significances and p-values of data were determined by nonparametric Kruskal–Wallis test followed by Dunn's post hoc test; *p≤0.05, **p≤0.01, ***p≤0.001, ****p≤0.0001. All data values are summarized in *Figure 6—source data 1*.

The online version of this article includes the following source data and figure supplement(s) for figure 6:

**Source data 1.** Quantification of neurotransmitter release parameters of STX1-null neurons transduced with STX1A^D231N, STX1A^R232N, STX1A^Y235R, STX1A^E238V, STX1A^V248K, STX1A^S249E, STX1A^D231N,R232N, and STX1A^V248K,S249E.

**Figure supplement 1.** Quantification of STX2 levels at the synapse.

**Figure supplement 1—source data 1.** Quantification of STX1A and Munc18-1 levels in STX1-null neurons transduced with STX1A^D231N, STX1A^R232N, STX1A^Y235R, STX1A^E238V, STX1A^V248K, STX1A^S249E, STX1A^D231N,R232N, and STX1A^V248K,S249E.

Finally, we examined the relationship between RRP size and release rate (*Figure 6I*) and PVR and release rate (*Figure 6H*). Changes that destabilize the primed state might be seen as a decrease in the RRP and an increase in the release rate. Changes in the height of the energy barrier for fusion may make it harder or easier for vesicles to fuse and could be indicated by a decrease or increase in the PVR, respectively, accompanied by a change in the release rate. Surprisingly, STX1A-2(SNARE), STX1A-2(Cter), and STX1A^D231N,R232N exhibit changes in both these correlations: decreased RRP and increased spontaneous release rate (*Figure 6I*), and increased PVR and spontaneous release rate (*Figure 6H*). This may suggest D231, R232 may have a role in the stability of primed vesicles and the energy barrier for fusion. However, the integrity of the SNARE domain of STX1A is more important than single-point changes, suggesting an additive effect of the residues of the C-terminal half of the SNARE domain.

## Discussion

In this study, we investigated the role and specificity of the SNARE domain of STX1 in regulating neurotransmitter release. We found that the C-terminus of the SNARE domain of STX1 is crucial in the regulation of multiple aspects of synaptic transmission, such as the clamping of spontaneous release and the formation and maintenance of the RRP of vesicles. Furthermore, it is involved in the regulation of speed and efficacy of Ca^2+-evoked release; however, these functions are dependent on the integrity of the full-SNARE domain of STX1 and regions outside of the SNARE domain.

### STX1 is fine-tuned for synaptic vesicle release

Current understanding of constitutive and regulated membrane fusion involves the idea of the energetically favorable interaction of four cognate SNARE domains that 'zipper-up' and pull the vesicle and plasma membrane together (*Fasshauer et al., 1998*). In the physiological context, vesicle fusion can vary dramatically in terms of efficacy and acceleration upon triggering, arguing for an extensive regulation machinery for the fusion process. An important role is attributed to regulatory proteins, such as complexins and synaptotagmins, that regulate various aspects of vesicle fusion, including synaptic vesicle docking and priming, clamping of spontaneous release, and speed and efficacy of neurotransmitter release (*Rizo, 2022*; *Rizo and Rosenmund, 2008*; *Südhof, 2013*). Although SNARE proteins are quite promiscuous in their assembly (*Bajohrs et al., 2005*; *Brunger, 2005*; *Fasshauer et al., 1998*; *Peng et al., 2013*; *Vardar et al., 2022*), variations in their SNARE motif sequence can lead to differential regulation of the fusion reaction beyond their role in zippering, for example, by modifying the surface charge (*Kádková et al., 2023*; *Ruiter et al., 2019*) or regulating interaction with modulatory proteins (*Schupp et al., 2016*; *Stepien et al., 2022*; *Zhou et al., 2015*). We show

that STX2 can rescue some aspects of neurotransmitter release in STX1-null neurons, supporting redundant functions among syntaxin isoforms in executing vesicle fusion. However, regulated fusion with non-cognate trans-SNARE complexes containing STX2 showed several unfavorable release properties, including slowed and inefficient $Ca^{2+}$-evoked release, a reduced pool of fusion-competent vesicles, while spontaneous release was greatly increased (*Figure 1*). Given these findings, and our previous findings which compared the release parameters of the tonic release isoform STX3 with STX1 (*Vardar et al., 2022*), we argue that STX1 is fine-tuned for phasic release with a high signal-to-noise ratio for evoked over spontaneous release.

## The C-terminus of the STX1 SNARE domain is important for the stability of the primed state of vesicles and the clamping of spontaneous fusion

As a general observation, the exchange of the whole or only the C-terminal half of the SNARE domains of STX1 and STX2 resulted in altered $Ca^{2+}$-dependent responses, PVR, PPR, short-term depression, as well as release rate (*Figures 2 and 3*). It is clear that the C-terminus of the STX1 SNARE domain is important for the stability of the primed state of the vesicles and the clamping of spontaneous fusion. While conducting a sub-saturating sucrose experiment (*Basu et al., 2007*; *Ruiter et al., 2019*; *Schotten et al., 2015*) could further confirm the hypothesis of reduced fusogenicity in STX1 chimeras, the small RRP obtained at 500 mM sucrose poses a limitation to our study. Therefore, we must rely on other parameters to interpret these results. Generally, changes in the molecular mechanisms for priming proportionally change the $Ca^{2+}$-evoked response while leaving the PVR unchanged. Changes in the PVR, PPR, release rate, and depression during high-frequency train stimulation, however, may indicate alterations in the fusogenicity of the vesicles that stem from aberrations in the mechanisms underlying the regulation of the fusion energy barrier (*Schotten et al., 2015*). In this light, we propose a model where the C-terminal half of the SNARE domain of STX1 plays a major role in the stabilization of the primed state and the clamping of the spontaneous release. Disrupting the C-terminus of the STX1A SNARE domain by adding that of the non-cognate STX2 (*Figure 7*, red dotted line) may impact vesicle fusogenicity by destabilizing the primed state and decreasing the fusion energy barrier that now becomes easier for vesicles to overcome (*Schotten et al., 2015*). Additionally, adding the C-terminus of the SNARE domain of STX1A to STX2 may increase the ability of the SNARE complex to stabilize the primed state by increasing the degree of the fusion energy barrier and thereby establish a clamp for spontaneous release (*Figure 7*, light green dotted line). This model supports our previous findings in which, by performing a similar chimeric analysis with STX3, we suggested the C-terminus of the SNARE domain of STX1 as a key regulator in the clamping of spontaneous vesicle fusion (*Vardar et al., 2022*). We discuss three possible mechanisms through which the C-terminus of the SNARE domain of STX1 might contribute to the stability of the primed state of synaptic vesicles and the clamping of spontaneous release.

## The interaction of the C-terminus of the SNARE domain of STX1A with Munc18-1 in the stabilization of the primed pool of vesicles

Previous studies have attributed the vesicle priming function to the N-terminal half of the SNARE domain (*Sørensen et al., 2006*; *Weber et al., 2010*) and proteins involved in the N-terminal nucleation of the SNARE complex, such as Munc13 and Munc18-1 (*Ma et al., 2011*; *Wang et al., 2020*). One line of thought proposes that these proteins prime vesicles to the plasma membrane by forming an SYB2-STX1 acceptor-complex for SNAP-25 (*Jiao et al., 2018*; *Lai et al., 2017*; *Parisotto et al., 2012*; *Stepien et al., 2022*; *Wang et al., 2019*). Thus, changes in Munc18-1 levels in STX1 and STX2 chimeras are one of the first hypotheses that come to mind as an underlying mechanism of the altered primed state of the vesicles. Notably, we did not observe changes in the levels of Munc18-1 when exchanging N-terminal halves of the SNARE domain between syntaxins. However, all the chimeras that contained the C-terminal half of the SNARE domain of the non-cognate isoform showed an increase in the levels of Munc18-1 compared to their native form (*Figures 4C and 5F*) while manifesting opposing effects in the stabilization of primed vesicles: STX2 chimeras showed a better stabilized primed state reflected in the increase of the RRP and a decrease in the fusogenicity parameters (*Figures 5H–J and 7*), whereas STX1A-chimeras showed signs of a destabilized primed state (*Figures 4F–H and 7*). These opposing phenotypes both accompanied by an increase in Munc18-1 levels might be the

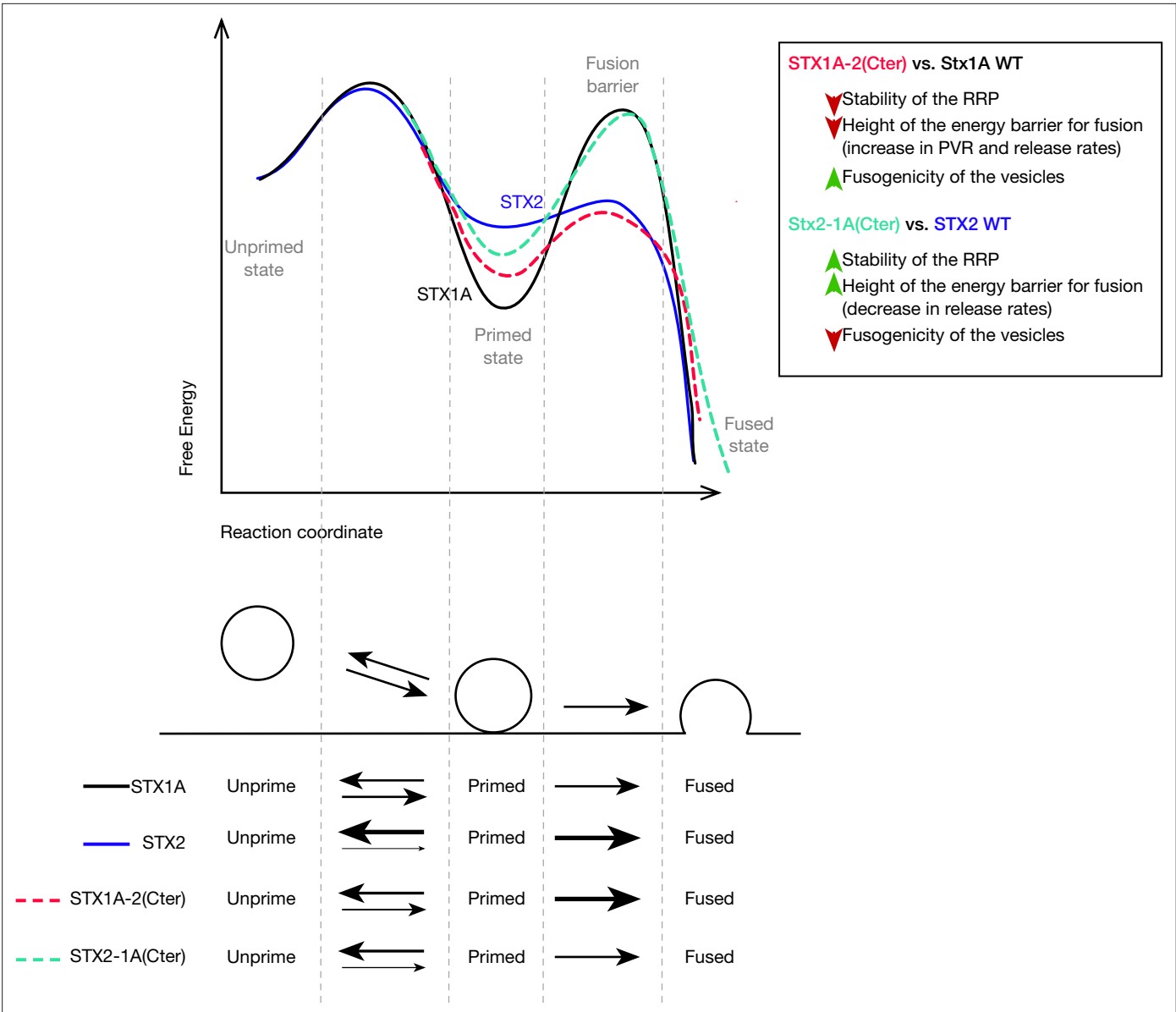

**Figure 7.** Speculative model based on the most important changes in electrophysiological properties found in STX1A-chimeras or STX2-chimeras compared to their WT isoforms. Energy landscape for priming and fusion of synaptic vesicles that have SNARE complexes formed with STX1A WT (black line), STX1A-2(SNARE or Cter) (red dotted line), STX2 WT (blue line), and STX2-1A(Cter) (light green dotted line). Summarized conclusions of our results based on the speculative model in. Changes in the equilibrium between the unprimed/primed state (reflecting the stability of the primed state) and the rate of fusion (energy landscape for fusion adapted from *Sørensen, 2009*).

manifestation of two modes of binding of Munc18-1 to the C-terminal half of the SNARE domain of STX1. First, Munc18-1 was shown to maintain an interaction with the C-terminus of the SNARE domain of STX1 during the N-terminal nucleation of the SNARE complex, important for the initiation of the SNARE complex assembly (*Stepien et al., 2022*). Additionally, the central cavity of Munc18-1 interacts with the C-terminal half of the SNARE domain of STX1 in its closed conformation, important for the chaperoning and trafficking of STX1 to the plasma membrane (*Han et al., 2011*, *Shi et al., 2011*). Mutations of the corresponding residues in STX1 (including D231 and E234), while affecting the binding and chaperoning functions of Munc18-1, did not affect the fusion in liposome fusion assays (*Shi et al., 2011*). We can speculate that the presence of the C-terminal half of the SNARE domain of STX2 might favor an open conformation of syntaxin due to an inability of Munc18-1 to chaperone the closed conformation and the open conformation of STX1 was previously shown to

display an increase in fusogenocity parameters (*Gerber et al., 2008*; *Zhou et al., 2013*; *Vardar et al., 2021*). On the other hand, the presence of the C-terminus half of the SNARE domain of STX1 allows for the binding of Munc18-1 and favors the chaperoning of a closed conformation of syntaxin, which would display reduced fusogenicity parameters. Therefore, it is tempting to speculate that Munc18-1 might have a qualitative effect on the vesicle stabilization rather than a quantitative one. In this case, it can be argued that an intact C-terminal half of the SNARE domain is a conditional requirement in order for Munc18-1 to fulfill the chaperoning and nucleation function on the N-terminus in a physiological setting, thus leading to a reduced RRP (*Figure 4F*).

## The participation of STX1A in the electrostatic regulation of the fusion energy barrier model

Besides the destabilization of the RRP, we found that the clamping of the spontaneous release is particularly susceptible to changes in single or double consecutive residues in the C-terminus of the SNARE domain (*Figure 6*). The degree of the vesicle fusion energy barrier is largely regulated by the electrostatic landscape of the fusion apparatus (*Shao et al., 1997*; *Trimbuch et al., 2014*; *Williams et al., 2009*), which will try to neutralize the negatively charged approaching membranes so that they can fuse. This is also influenced by the total charge of the SNARE domain complex (*Ruiter et al., 2019*). Thus, changes in the net charge of the SNARE complex can cause altered vesicle fusogenicity, and as a result, altered release rates at the synapse (*Ruiter et al., 2019*). The observation of SNAP-25 charge-reversal mutations proposed a model by which a positive net charge increase lowers the fusion energy barrier and increases fusion rates because it neutralizes the negative charges of the membranes, while a negative net charge increase increases the negative electrostatic fusion energy barrier, causing a decrease in fusion rates (*Kádková et al., 2023*; *Ruiter et al., 2019*). In this light, it is plausible to argue that the differences in the release behavior led by the single- or double-point mutations in the C-terminal half of the SNARE domain of STX1A (*Figure 6*) might be due to the altered electrostatic net charge. The net charge difference in the C-terminal half of the SNARE domain between STX1A and STX2 is only one positive charge that would only lead to a minor change in the release rate. However, the release rates we observed are 15-fold (in the native form) (*Figure 1K*) or 17-fold and 8-fold (in STX1A-chimeras) (*Figure 2H*) increased. Additionally, the mutations STX1A$^{D231N,R232N}$ and Stx1A$^{V248K,S249E}$ that result in a null net charge difference increased the spontaneous release rate four-fold compared to STX1A WT or mutation STX1A$^{S249E}$ that should reduce the release rate according to this hypothesis as it possesses one negative net charge difference increases the spontaneous release rate threefold (*Figure 6G*). Therefore, our results suggest that the changes in release rate in STX1A point mutations are not caused merely by electrostatic changes but by an independent mechanism from the electrostatic model of regulation of release.

## The interaction of STX1A with the C2B domain of SYT1 through the primary interface

What can be the mechanism through which the C-terminal half of the SNARE domain regulates vesicle fusion if it is not solely through electrostatic changes? Interestingly, neurons expressing STX2 showed a slowed-down Ca$^{2+}$-evoked release, a dramatic increase in the spontaneous release rate, and a decrease in the RRP (*Figure 1*). Additionally, STX1A-chimeras that contained the C-terminus of the SNARE domain of STX2 showed a decrease in the RRP and a dramatic unclamping of the spontaneous release rate (*Figure 2*). This is reminiscent of the well-established phenotype of SYT1-null mouse neurons (*Bouazza-Arostegui et al., 2022*; *Stepien et al., 2022*; *Xu et al., 2007*; *Xue et al., 2009*) and suggests a possible alteration in the function of SYT1 in these neurons. Moreover, mutating individual residues in the C-terminal half of the SNARE domain that differ between STX1A and STX2 showed a particular effect on the clamping of spontaneous release and the RRP in the case of D231 and R232 (*Figure 6E*), two functions associated to SYT1-SNARE interaction (*Schupp et al., 2016*). Some of these residues (D231 and E238) have been identified by crystal structure studies to be involved in the putative interaction between the C2B domain of SYT1 and the C-terminal half of the SNARE domain of STX1 and SNAP-25, referred to as the primary interface (*Zhou et al., 2015*). Mutations of the residues at the primary interface in SYT1 and SNAP-25 identified this putative interaction as a key regulation site for the clamping of spontaneous release and Ca$^{2+}$-triggered release (*Chang et al., 2018*; *Schupp et al., 2016*; *Zhou et al., 2015*). The partial rescue of the speed of

$Ca^{2+}$-evoked release, the RRP, and the clamping of the spontaneous release in STX2 chimeras that contained the C-terminus of the SNARE domain of STX1A (*Figure 3*) suggested a gain of function through the interaction with SYT1. However, none of the changes in the C-terminus of the SNARE domain of STX1A resulted in a change in the speed of release (*Figures 2 and 6*), and the release rates obtained in STX1A-2(SNARE) and STX1A-2(Cter) were dramatically higher (*Figure 2*) than those in SYT1-KO neurons (*Bouazza-Arostegui et al., 2022*). In this light, our data support the primary interface as a key site for the regulation of spontaneous release, in addition to further unidentified mechanisms involving other residues in the C-terminal half of the SNARE domain of STX1A. Notably, it argues against STX1 as part of the $Ca^{2+}$-evoked release mechanisms through its interaction with the C2B domain of SYT1, supporting our previous findings that identified regions outside of the SNARE domain of STX1, such as the JMD and TMD, as regulators of the synchronization of neurotransmitter release (*Vardar et al., 2022*).

Finally, it is possible that the changes in the spontaneous release rate and the priming stability may stem from a reduced stability of the SNARE complex itself through putative interactions between outer surface residues. Studies of the kinetics of assembly of the SNARE complex that mutate solvent-accessible residues in the C-terminal half of the SNARE domain of SYB2 have shown reduction in the stability of the SNARE complex assembly and are correlated with impaired fusion (*Jiao et al., 2018*). However, STX1 mutations of outward residues were inconclusive and were always accompanied by hydrophobic layer mutations (*Jiao et al., 2018*), which affect the assembly kinetics and energetics of the SNARE complex (*Ma et al., 2015*). Single-molecule optical-tweezer studies have focused on the impact of regulatory molecules on the stability of assembly such as Munc18-1 (*Ma et al., 2015*; *Jiao et al., 2018*) and complexin (*Hao et al., 2023*), or on the intrinsic stability of the hydrophobic layers in the step-wise assembly of the SNARE complex (*Gao et al., 2012*; *Ma et al., 2015*; *Zhang, 2017*). Although the conserved hydrophobic layers in the SNARE domains of STX1A and STX2 (*Figure 1*) suggest unchanged zippering and intrinsic stability of the complex, further studies addressing the contribution of surface residues on the stability of the alpha-helical structure of the SNARE domain of STX1 (*Li et al., 2022*) or the stability of the SNARE complex should be conducted.

## Materials and methods

### Animal maintenance and generation of mouse lines

All animal-related procedures and experiments were performed in accordance with the guidelines and approved by the animal welfare committee of Charité-Universitätsmedizin and the Berlin state government Agency for Health and Social Services under license number T0220/09. To generate the STX1-null mouse model, we bred the conventional *Stx1a*-knockout (KO) line, in which exons 2 and 3 were deleted (*Stx1a*(-/-)) (*Gerber et al., 2008*), with the *Stx1b* conditional KO line in which exons 2–4 were flanked by loxP sites (*Stx1b*(*flox/flox*)) (*Wu et al., 2015*).

### Neuronal cultures

Primary hippocampal neuronal cultures were prepared from STX1-null mice at postnatal days 0–1 and were seeded onto a continental astrocyte feeder layer (for immunocytochemistry, 40K neurons/well in 12-well plates) or astrocyte micro-dot islands (for electrophysiology, 4K neurons/well in 6-well plates) as previously described (*Vardar et al., 2016*; *Xue et al., 2007*). Cortical neuronal cultures were prepared from the same mice and plated onto continental astrocyte feeder layer for western blot experiments (500K neurons/well in 6-well plates). Cultures were incubated in NeuroBasal-A (NBA) medium (Invitrogen) supplemented with B-27 (Invitrogen), 50 IU/ml penicillin, and 50 µg/ml streptomycin at 37°C before and during the experiments. Neuronal cultures were incubated for DIV13–20.

### Lentiviral construct production and viral infection

Point mutations were introduced into *Stx1a* or *Stx2* using a TA cloning vector with the QuikChange Site-Directed Mutagenesis Kit (Stratagene). Chimeric constructs of both syntaxin isoforms were obtained using Gibson Assembly (NEB). For the viral infection, the cDNA of mouse *Stx1a* (NM_016801.4), *Stx2* (NM_007941.2), chimeric, and point-mutation constructs were cloned into a lentiviral shuttle vector containing a nuclear localization signal (NLS) GFP-P2A expression cassette under the human synapsin-1 promoter. For obtaining the *Stx1b*-KO in *Stx1a*(-/-);*Stx1b*(*flox/flox*) neurons, we used a

lentiviral construct that induces the expression of improved Cre-recombinase (iCre) fused to NLS-RFP-P2A under the control of human synapsin-1 promoter. All viral particles were made by the Viral Core Facility of Charité-Universitätsmedizin as previously described (*Lois et al., 2002*). Empty NLS-GFP-P2A constructs were used as control. For all experiments, cultures were infected with lentiviral particles at DIV1–2.

## Neuron viability

For neuron survival assays, we quantified the number of STX1-null hippocampal neurons transduced with *Stx1a, Stx2,* and GFP (as control) at DIV 15, 22, and 29, and compared this to the number of neurons at DIV8. Also, 2 wells per group were plated and 15 random ROIs of 1.23 mm$^2$ per well were imaged at the different time points (30 total ROIs per group at each time point per culture). Phase-contrast bright-field images and fluorescent images with excitatory lengths of 488 and 555 nm were acquired on a DMI 400 Leica microscope, DFC 345 FX camera (Leica), HCX PL FlUPTAR ×10 objectives (Leica) and LASAD software (Leica). The neurons were counted offline with the 3D Object Counter function in Fiji software (*Vardar et al., 2016*). For the example images, the cultures were fixed at the corresponding time points in 4% paraformaldehyde (PFA) in 0.1 M phosphate-buffered saline (PBS), pH 7.4 , and immunocytochemistry was done as follows.

## Immunocytochemistry and image acquisition

High-density hippocampal neuron cultures (40 × 10$^3$) were plated onto the astrocyte-feeder layer and infected at DIV1–2. At DIV14–15, the cultures were fixed in 4% PFA in PBS for 10 min, permeabilized in 0.1% Tween-20 PBS (PBS-T) for 45 min at room temperature (RT), and blocked in 5% normal donkey serum in PBS-T for 1 hr at RT. To detect our proteins of interest, the neurons were incubated with the primary antibodies in PBS-T at 4°C overnight (O/N). For protein expression analysis, neurons were incubated with guinea pig polyclonal anti-VGlut1 (1:4000; Synaptic Systems), mouse monoclonal anti-STX1A (1:1000; Synaptic Systems), and rabbit polyclonal anti-STX2 (1:1000; Synaptic Systems) or rabbit polyclonal Munc18-1 (1:1000; Sigma-Aldrich). Secondary antibodies conjugated with rhodamine red, or Alexa Fluor 488, or 647 (1:500; Jackson ImmunoResearch) in PBS-T were applied for 1 hr at RT in the dark. For the example images for survival assay, neurons were incubated with chicken polyclonal anti-MAP2 (1:2000; Merck Millipore) and secondary antibody Alexa Fluor 488. The coverslips were mounted on glass slides with Mowiol mounting agent (Sigma-Aldrich). For quantitative analysis, every comparable group in each culture was treated with the same antibody solution. Images were acquired with an Olympus IX81 epifluorescence microscope with MicroMax 1300YHS camera (Princeton Instruments) and MetaMorph software (Molecular Devices). Images for the example images of the survival assay and the quantitative analysis of protein expression were taken with an optical magnification of ×10 and ×60, respectively. The exposure times of each wavelength were kept constant for all experimental groups in one culture. Overexposure and photobleaching was avoided by monitoring the fluorescent saturation level at the synapse. To image glutamatergic neurons, 7–9 images that had a fluorescent signal for VGlut1 staining were taken from each group in each experimental replicate. Data was analyzed offline on ImageJ. Then, 3–5 regions per image were selected, which showed neuronal projections marked by VGlut1-positive labeling. Vglut1-positive puncta were selected using MaxEntropy threshold function of the selected regions, creating a VGlut1-positive ROI. The intensities of VGlut1, STX1A, STX2, and Munc18-1 were measured using the corresponding ROIs. For the quantification of relative protein expression at glutamatergic synapses, the ratio between STX1A, STX2, or Munc18-1 to VGlut1 was calculated in each selected region. To compare between groups, the data was normalized to the STX1A group or STX2 group.

## Electrophysiology

To assess synaptic function, whole-cell patch-clamp recordings in autaptic neurons were performed at DIV13–19 at RT. Synaptic currents were recorded using a MultiClamp 700B amplifier (Molecular Devices), and data was digitally sampled at 10 kHz and low-pass filtered at 3 kHz with an Axon Digidata 1440A digitizer (Molecular Devices). Data acquisition was controlled by Clampex10 software (Molecular Devices). Series resistance was compensated at 70%, and only neurons with a series resistance lower than 10 MΩ were further recorded. Neurons were continuously perfused using a fast perfusion system (1–2 ml/min) with an extracellular solution (unless stated otherwise) that contains

140 mM NaCl, 2.4 mM KCl, 10 mM HEPES, 10 mM glucose, 2 mM CaCl$_2$, and 4 mM MgCl (295–305 mOsm, pH 7.4). Borosilicate glass pipettes were pulled, yielding a resistance between 2 and 5 MΩ. Pipettes were filled with a KCl-based intracellular solution containing 136 mM KCl, 17.8 mM HEPES, 1 mM EGTA, 4.6 mM MgCl$_2$, 4 mM Na$_2$ATP, 0.3 mM Na$_2$GTP, 12 mM creatine phosphate, and 50 Uml1 phosphocreatine kinase (300 mOsm; pH 7.4). Only neurons that express RFP and GFP were selected for recording, and cells with a leak current higher than 300 pA were excluded. Neurons were clamped at –70 mV during all protocols of electrophysiological recording. For triggering action potential (AP), evoked release neurons were stimulated by a 2 ms depolarization to 0 mV and EPSCs were registered. To measure the synchronicity and kinetics of the AP-evoked responses, we inverted the EPSC change and integrated our signal. For measuring spontaneous release (mEPSC), we recorded at –70 mV for 48 s and for 24 s in the presence of the AMPA-receptor antagonist NBQX (3 µM) (Tocris Bioscience) diluted in extracellular solution. To calculate the frequency of mEPSC, we filtered at 1 kHz and analyzed using a template-based miniature event detection algorithm implemented in the AxoGraph X software. We then subtracted the measured mEPSC in NBQX to the mEPSCs measured in extracellular solution. The release of the RRP of synaptic vesicles was triggered by the application of a 500 mM hypertonic sucrose solution diluted in extracellular solution for 5 s (*Rosenmund and Stevens, 1996*), and the RRP size was estimated by integrating the area of the sucrose-evoked current setting as baseline the steady-state current. The vesicular release probability (PVR) of each neuron was determined by the ratio between the charge of the EPSC and the size of the RRP. The spontaneous release rate was calculated as the ratio between the mEPSC frequency and the number of primed vesicles. This determines the fraction of the RRP, which is spontaneously released per second. To measure the cumulative charge transfer of the synaptic responses, we inverted the EPSC charge and integrated the signal. Short-term plasticity was examined either by evoking 2 AP with 25 ms interval (40 Hz) or a train of 50 AP at an interval of 100 ms (10 Hz). Data were analyzed offline using Axograph10 (Axograph Scientific).

## Western blot

Protein lysates from cortical mass cultures were prepared by lysing neurons in 200 µl lysis buffer containing 50 mm Tris/HCl, pH 7.9, 150 mm NaCl, 5 mm EDTA, 1% Triton X- 100, 0.5% sodium deoxycholate, 1% Nonidet P-40, and 1 tablet of cOmplete Protease Inhibitor (Roche) for 30 min on ice. Samples were boiled for 5 min at 95°C. Equal amounts of protein were loaded onto a 12% SDS-PAGE and run in electrophoresis buffer at 80 V for 30 min and then 120 mV for 1 hr. Proteins were transferred onto a nitrocellulose membrane at 50 mA O/N. The membranes were blocked in 5% milk for 1 hr at RT and incubated with the corresponding primary antibody diluted in PBS for 1 hr at RT. Membranes were incubated with mouse monoclonal anti-STX1A (1:10,000; Synaptic Systems), rabbit polyclonal anti-STX2 (1:10,000; Synaptic Systems), rabbit polyclonal anti-STX3A (1:10,000; Synaptic Systems), rabbit polyclonal anti-Munc18-1 (1:10,000; Sigma-Aldrich), and mouse monoclonal anti-βTubIII (1:10,000; Sigma) (as loading control). Secondary antibodies conjugated with HRP-conjugated goat secondary antibodies (1:10,000; Jackson ImmunoResearch) diluted in PBT were applied for 1 hr at RT. Membranes were then incubated with ECL Plus Western Blotting Detection Reagents (GE Healthcare Biosciences) and luminal signal was visualized and imaged in Fusion FX7 image and analytics system (Vilber Lourmat).

## Statistical analysis

Data presented in box-whisker plots were compiled as single observations, median, quartiles, and outliers. Data in bar graphs and X-Y plots present means ± SEM. All data were tested for normality with D'Agostino–Pearson test. Data from two groups with nonparametric distribution were subjected to Mann–Whitney test. Data from two groups with normal distribution were subjected to unpaired two-tailed *t*-test. Data from more than two groups were subjected to Kruskal–Wallis followed by Dunn's post hoc test when at least one group showed a nonparametric distribution. Data from more than two groups were subjected to ordinary one-way ANOVA when all groups showed a parametric distribution. STP was subjected to two-way ANOVA. Statistical analyses were performed using Prism 7 (GraphPad). All statistical data are summarized in the corresponding 'Source Data' tables. Fold increase/decrease and percentage increase/decrease were done using the mean values of each parameter.

## Acknowledgements

We thank the Charité Viral Core facility, Katja Pötschke, and Bettina Brokowski for virus production, Berit Söhl-Kielczynski and Heike Lerch for technical assistance, Melissa Herman for their contribution to the manuscript, and all the Rosenmund Lab members for the discussion. This project was funded by the German Research Council (DFG) grants 399894546, 184695641, 278001972, and 390688087 and an SFB958 doctoral fellowship.

## Additional information

### Funding

| Funder | Grant reference number | Author |
|---|---|---|
| Deutsche Forschungsgemeinschaft | 399894546 | Andrea Salazar Lázaro Thorsten Trimbuch Gülçin Vardar Christian Rosenmund |
| Deutsche Forschungsgemeinschaft | 436260754 | Andrea Salazar Lázaro Thorsten Trimbuch Gülçin Vardar Christian Rosenmund |
| Deutsche Forschungsgemeinschaft | 388271549 | Andrea Salazar Lázaro Thorsten Trimbuch Gülçin Vardar Christian Rosenmund |
| Deutsche Forschungsgemeinschaft | 278001972 | Andrea Salazar Lázaro Thorsten Trimbuch Gülçin Vardar Christian Rosenmund |
| Deutsche Forschungsgemeinschaft | 184695641 | Andrea Salazar Lázaro |

The funders had no role in study design, data collection and interpretation, or the decision to submit the work for publication.

### Author contributions

Andrea Salazar Lázaro, Data curation, Formal analysis, Investigation, Writing - original draft; Thorsten Trimbuch, Conceptualization, Resources; Gülçin Vardar, Conceptualization, Supervision, Writing - review and editing; Christian Rosenmund, Conceptualization, Supervision, Funding acquisition, Writing - original draft, Project administration, Writing - review and editing

### Author ORCIDs

Andrea Salazar Lázaro https://orcid.org/0009-0008-3114-8282
Thorsten Trimbuch https://orcid.org/0000-0001-7512-8955
Christian Rosenmund https://orcid.org/0000-0002-3905-2444

### Ethics

All animal-related procedures and experiments were performed in accordance with the guidelines and approved by the animal welfare committee of Charité-Universitätsmedizin and the Berlin state government Agency for Health and Social Services, under license number T0220/09.

Reviewer #1 (Public Review): https://doi.org/10.7554/eLife.90775.3.sa1
Reviewer #2 (Public Review): https://doi.org/10.7554/eLife.90775.3.sa2
Reviewer #3 (Public Review): https://doi.org/10.7554/eLife.90775.3.sa3
Author Response https://doi.org/10.7554/eLife.90775.3.sa4

### Data availability

All data generated or analysed during this study are included in the manuscript and supporting files.

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
