## [Editor Report · eLife assessment]

This **important** study presents a series of results to uncover the role of C-terminal half of the Syx1 SNARE domain. The evidence supporting the conclusions is **convincing**. The paper will be of broad interest to biophysicists and neurobiologists.

---

## [Referee Report · Reviewer #1 (Public Review)]

In this systematic and elegant structure-function analysis study, the authors delve into the intricate involvement of syntaxin 1 in various pivotal stages of synaptic vesicle priming and fusion. The authors use an original and fruitful approach based on the side-by-side comparison of the specific contributions of the two isoforms syntaxin 1 and syntaxin 2, and their respective SNARE domains, in priming, spontaneous and Ca2+-dependent glutamate release. The experimental approach, mastered by the authors, offers an ideal means of unraveling the molecular roles played by syntaxins. Although it is not easy to come up with a model explaining all the observed phenotypes, the authors carefully restrict their conclusions to the role of the C-terminal half of the syntaxin1 C-terminal SNARE domain in the maintenance of the RRP and the clamping of neurotransmitter release. The study is carefully carried out, the conclusions are supported by high quality data and the manuscript is clearly written. In addition, the study clearly set new questions than open new paths for future experimental work.

---

## [Referee Report · Reviewer #2 (Public Review)]

Summary:

The manuscript by Salazar-Lázaro et al. systematically dissects out the different functional properties of the SNARE-domains of syntaxin-1 and syntaxin-2. By systematically substituting the SNARE-domain (or its C- or N-terminal half) into the non-cognate counterpart, the authors find that the C-terminal half of the SNARE-complex is especially important for maintaining RRP size and clamping spontaneous release. They also mutate single residues, to further nail down the effect. Overall, this is an interesting manuscript, which sheds light on the functionality of different co-expressed SNARES.

Strengths:

The strength of the manuscript is the systematic dissection, using substitution of either SNARE-domain into the other syntaxin, together with the state-of-the art methods. The authors follow up with a substitution of single and paired residues. This is a large undertaking, which has been very well carried out.

Weaknesses:

No major weaknesses. The large number of experiments paint a somewhat complicated picture because the process under study is complicated.

---

## [Referee Report · Reviewer #3 (Public Review)]

Summary:

In this manuscript, Salazar-Lázaro et al. presented interesting data that C-terminal half of the Syx1 SNARE domain is responsible for clamping of spontaneous release, stabilizing RRP, and also Ca2+-evoked release. The authors routinely utilized the chimeric approach to replace the SNARE domain of Syx1 with its paralogue Syx2 and analyzed the neuronal activity through electrophysiology. The data are straightforward and fruitful. The conclusions are reasonable.

Strengths:

The electrophysiology data that illustrate the important functions of Syx1 in clamping of spontaneous release, stabilizing RRP, and also Ca2+-evoked release were clear and convincing.

Weaknesses:

One weakness is that the authors did not go deep into the underlying molecular mechanisms experimentally, either because of a variety of complicated possibilities or limited space of the manuscript.

---

## [Author Response]

The following is the authors’ response to the original reviews.

We wish to thank the reviewers for their helpful insightful comments. Their concerns were mainly related to the interpretation of the data, help in clarifying our statements and improving our discussion.

**Reviewer #1 (Recommendations For The Authors):**
This is a very interesting study It involves the utilization of hippocampal neuronal cultures from syntaxin 1 knock-out mice. These cultures serve as a platform for monitoring changes in synaptic transmission through electrophysiological recording of postsynaptic currents, upon lentiviral infection with various isoforms, chimeras, and point mutations of syntaxins.The authors observe the following:(1) Syntaxin2 restores neuronal viability and can partially rescue Ca2+-evoked release in syntaxin1 knock-out neurons that it is much slower (cumulative charge transfer differences) and with a clearly smaller RRP than when rescued with syntaxin1. In contrast, syntaxin2-mediated rescue leads to a high increase in spontaneous release (Figure 1). Convincingly, the authors conclude that syntaxin 1 is optimized for fast phasic release and for clamping of spontaneous release, in comparison with syntaxin2.(2) The replacement of the SNARE domain (or its C-terminal part) of syntaxin1 by the SNARE domain of syntaxin2 (or its C-terminal part) rescues the fast kinetics, but not the amplitude, of Ca2+-evoked release. This is associated with a decrease in the size of the RRP and an increase in spontaneous release. The probability of vesicular release (PVR) is a little bit increased, which is intriguing because a little decrease would be expected instead according to the reduced RRP, indicating that an enhancement of Ca2-dependent fusion is occurring at the same time by unknown mechanisms as the authors properly point out. The replacement of the Analogous experiments in which the SNARE domain of syntaxin1 is replaced into syntaxin2, reveals the exitance of differential regulatory elements outside the SNARE domain.(3) Different constructs of syntaxin 1 and syntaxin 2 display different expression levels. On the other hand, the expression levels of Munc-18 are associated with the characteristics of the transfected specific syntaxin construct. In any case, the electrophysiological phenotypes cannot be consistently explained by changes in Munc-18.(4) Mutations in several residues of the outer surface of the C-terminal half of the syntaxin1 SNARE domain lead to alterations in the RRP and the frequency of spontaneous release, but the changes cannot attributed to a change in the net surface charge, because the alterations occur even in paired mutations in which electrical neutrality is conserved.Comments:(1) This is a comment regarding the interpretation of the results. In general, the decrease in the RRP size is associated with the increased frequency of spontaneous release due to unclamping. The authors claim that both phenomena seem to be independent of each other. In any case, how can the authors discard the possibility that the unclamping of spontaneous release leads to a decrease in the RRP size?

The main argument against the reduction of the RRP being caused by the observed increase in the mEPSC frequency is based on kinetics of refilling and depletion. The average time a vesicle fuses spontaneously after it becomes primed is 500 – 1000 seconds (spontaneous vesicle release rate – STX1 Figure 1, Figure 2 and Figure 3). The time it takes to refill the RRP after depletion is in the order of 3 seconds (Rosenmund and Stevens, 1996). Therefore, the refilling of the RRP is more than 100 times faster. Even when the spontaneous release would increase 5 fold, this would lead to less than 5 % of the steady state depletion of the RRP.

(2) The authors have analyzed the kinetics of mEPSCs and found differences (Fig2-Supp. Fig1; Fig2-Supp. Fig1). It would be interesting and pertinent to discuss these data in the context of potential phenotypes in the fusion pore kinetics involving syntaxin1 and syntaxin2 and their SNARE domains. Indeed, the figure will improve by including averaged traces of mEPSCs.

We thank the reviewer for the idea. Upon closer examination of the changes in mEPSC rise time and mEPSC decay time we noticed a minor slowing in the mEPSC rise time from 0.443ms (SEM±0.0067) of STX1A to 0.535ms (SEM±0.0151) for STX1A-2(SNARE) or 0.507ms (SEM±0.01251) for STX1A-2(Cter), while the mEPSC half widths did not change significantly. It is possible that the measured change is related to the detection algorithm as mEPSC detection at elevated frequencies becomes more difficult due to increased overlap of event, and we therefore prefer to refrain from making any mechanistic claims.

Minor comments:(1) Fig2 J; Fig 3 J. It is difficult to distinguish between different colors and implementing a legend within the graph will be very helpful.(2) Fig3 H. Please change the color of the box plot for Stx1 A to improve the contrast with the individual data points.(3) Page 6. Line 225. "Figure 2D and E" should be corrected to "Figure 2C and D"

(1) Colors were changed for clearer visualization. (2) Unfortunately, changing the color did not improve the contrast with the individual plots. However, the numerical data is all included in the data sheets of the corresponding figure. (3) The mistake was corrected.

**Reviewer #2 (Recommendations For The Authors):**
Line 135-136: Are cited numbers cited in the text mean and SEM? Please indicate.Line 139 and Figure 1G: The difference between purple and blue was very hard to see on my hard copy.Line 152: Reference to Figure 1L should probably be 1K.Line 183: Reference to Figure 2C should probably be Figure 2F.Line 225: Reference to Figure 2D and 2E should probably be 2C and 2D.Line 239: Reference to Figure 3I should probably be 3H.

All typos were addressed and colors were changed for better visualization.

Line 210-211: Sentence ("One of the benefits..") is hard to understand.

Thank you for noticing this mistake, agreeably the the sentence did not add any important or new information and so it was deleted. Additionally, the message of the mentioned sentence was already clearly stated in lines 209-211.

Figure 4E-H misses data for STX2, for the figure to be arranged like Figure 5.

Given that STX1 is the endogenous syntaxin in hippocampal neurons, we use it at a control for all the analysis done in STX2 and STX2-chimera experimental groups, thus it is included in Figure 3 and 5.

It appears that the authors do not present or discuss the Western Blot in Fig. 4D. Are the quantitative results of the Western Blot consistent with or different from the quantification of the immunostainings (Fig. 4B-C)? A similar question for Figure 5D, which also seems not to be presented.

In terms of quantification, we have relied mainly on the ICC experiments because they test also for putative impairments in transport to the presynaptic compartment. Our WB data are overall consistent with the results, but were not used to quantitate expression of our syntaxin chimeras and mutations in the STX1-null hippocampal neuron model.

Figure 6F-G: The normalization of spontaneous vesicular release rates is not clear, because the vesicular release rates already contain a normalization (mEPSC rate divided by RRP size). Is a further normalization of the STX1A condition informative? The authors should consider presenting the release rates themselves. In any case, the normalization should be presented/explained, at least in the legends.

The reviewer is in principle correct. Due to the large number of experimental groups we had to perform recordings from multiple cultures, where not all experimental groups were present, while the WT STX1 was present as a consistent control. The reduce culture to culture variability, additional normalization to the WT control group was performed. However, we also included the raw data numerical values in the data-source sheets (Normalized and absolute), which produce a similar overall outcome.

References to Figure 7 subpanels (A, B, and C) are missing.

Thank you for the comment. We have integrated all panels into one for better representation and understanding since they are representative of one another.

Lines 330-339 and Figure 7 in Discussion: the authors discuss that adding the non-cognate STX2 SNARE-domain to syntaxin-1 might destabilize the primed state and decrease the fusion energy barrier (as indicated in Figure 7C). What is the evidence that the decrease in RRP size is not caused solely by the depletion of the pool due to the increased spontaneous fusion?

Please see the comments to major point 2 of reviewer 1.

Statistics: Missing is the number of observations (n) for all data. Even if all data points are displayed, this should be stated.

N numbers are included in the data sheets attached to each figure.

The statement (start of Discussion,) that the SNARE-domain of STX1 'plays a minimal role in the regulation for Ca2+-evoked release' is somewhat puzzling, since without the SNARE-domain in STX1 there would be no Ca2+-evoked release. I guess these statements (similar statements are found elsewhere) are due to the interesting finding that STX2 leads to a decrease in release kinetics, compared to STX1, and this is not (entirely) due to differences in the SNARE-domain. I would suggest rephrasing the finding in terms of release kinetics. Also, the statement in the last sentence of the Abstract is not clear.

Thank you for pointing this out and we agree that our experiments showed strong impact of the syntaxin isoform exchange on release kinetics and overall release output. A similar comment came also from reviewer #3 and so, we have addressed both comments as one.

Our confusing statement resulted from the order of the presented results and our summarizing remarks for each section. Our statement reflected our finding that mutating residues in the C-terminal part of the STX1 SNARE motif affected only spontaneous release and RRP size but not release efficacy. We now state (pg. 6 lines 231-233) that the data observed from the comparison of “the results obtained from the Ca2+-evoked release between STX1 and STX2 support major regulatory differences of the domains outside of the SNARE domain between isoforms”.

We have changed the abstract pg. 2 lines 55-56

We have changed the introduction pg. 3 lines 102-105 for a better contextualization.

We have changed the start of the discussion pg. 9 lines 250-252 for better contextualization.

**Reviewer #3 (Recommendations For The Authors):**
In this manuscript, Salazar-Lázaro et al. presented interesting data that C-terminal half of the Syx1 SNARE domain is responsible for clamping of spontaneous release, stabilizing RRP, and also Ca2+-evoked release. The authors routinely utilized the chimeric approach to replace the SNARE domain of Syx1 with its paralogue Syx2 and analyzed the neuronal activity through electrophysiology. The data are straightforward and fruitful. The conclusions are partly reasonable. One obvious drawback is that they did not explore the underlying mechanism. I think it is easy for the authors to carry out some simple assays to verify their hypothesis for the mechanism, instead of just talking about it in the discussion section. In all, I appreciate the data presented in the manuscript. If the authors could supply more data on the mechanisms, this would be important research in the field. Some critical comments are listed below:

We thank the reviewer for his/her comments and suggestions.

Major comments:(1) In pg.3, lines 102-104, the authors stated that 'We found that the C-terminal half of the SNARE domain of STX1.. ..while it is minimally involved in the regulation of Ca2+-evoked release.' But in pg.5, lines 174-176, they wrote that 'Replacement of the full-SNARE domain (STX1A-2(SNARE)) or the C-terminal half (STX1A-2(Cter)) of the SNARE domain of STX1A with the same domain from STX2 resulted in a reduction in the EPSC amplitude (Figure 2B).' and in pg.5-6, lines 197-199, they wrote that 'Taken together our results suggest that the C-terminal half of the SNARE domain of STX1A is involved in the regulation of the efficacy of Ca2+-evoked release, the formation of the RRP and in the clamping of spontaneous release.' It puzzles me a lot as to what the authors are really trying to express for the relationship between C-half of the SNARE complex and Ca2+-evoked release (i.e., minimally involved or significantly participate in the process?). Please clarify and reorganize the contexts.

Please see our reply to the last comment of reviewer 2.

(2) Figure 1-figure supplement 1, the authors should analyze Syx1/VGlut1 level additionally. And, if possible, compare the difference between Syx1/VGlut1 and Syx2/VGlut1.

The levels of STX1/VGlut1 and STX2/VGlut1 were analyzed in detail in Figures 4 and 5.

The direct comparison between the expression levels of these two proteins is not possible since affinities of the antibodies to the target proteins are different and can induce potential biases. While this could be overcome by the use of a FLAG-tag to the syntaxin proteins, we have not utilized this approach in this publication. We in addition inferred sufficient and comparable expression of both syntaxins from their ability to rescue some of syntaxin1 loss of function phenotypes.

(3) Figure 2D only analyzed the EPSC half-width, could the author alternatively analyze the rise/decay time? Also, in Figure 3-figure supplement 1, does it refer to the kinetic parameters of Syx2-1A in Figure 3? It is very confused.

We have changed the text accordingly and each parameter is referenced to its corresponding figure for clarity. As for the decay and rise time of STX1 and STX1-chimeras, they are in Figure 2-figure supplement 1A and B.

(4) On pg.4, lines 151-152, 'Finally, no change was observed in the paired-pulse ratio (PPR) between STX1A and STX2 groups (Figure 1L).' does not contain any explanations and comments for this observation in the texts.

The small EPSC amplitudes and altered kinetics on the STX2 constricts (Figure 1 and Figure 3) have made it more difficult to quantitate paired pulse experiments. Therefore, we preferred not to overinterpret these measurements. The findings that the paired pulse data were not significantly different, fit with the vesicular release probability measurements which showed no major changes. We have made our statement on this basis.

(5) On pg.6, lines 235-236, the authors wrote that 'Additionally, we found that only STX2-1A(SNARE) and STX2-1A(Cter) could rescue the RRP to around double of what we measured from STX2 and STX2-1A(Nter) (figure 3F)'. However, in Figure 3F, the authors indicated 'n.s.' (p>0.05) for the differences between STX2 and STX2-1A(SNARE)/STX2-1A(Cter). It is perplexing how the authors interpret their data. Definitely, the p-value could not be arbitrarily used as a criterion of difference. An easier way is that indicating the exact p-values for each comparison (indicate in figure legends or list in tables).

We apologize for any confusion, and hope the modification gives more clarity in our interpretation. The calculated p-values are included in attached data source tables and hope this will provide clarity to our comparative analysis. We have changed the text in pg 7 lines 238-241 and are cautious to overinterpret these results and rely more on the data observed in STX1A-chimeras, which show significant changes in the RRP.

(6) I noticed that the authors preferred using 'xx% increase/decrease' or 'xx-fold increase/decrease' to interpret their inter-group data. I would doubt whether the interpretations are appropriate. First, it seems that most of the individual scatters from one set were not subject to Gaussian distribution; also, the authors utilized non-parameter tests to compare the differences. Second, the authors did not explicitly indicate the method to calculate the % or fold, e.g., by comparing mean value or median. I think it is a bad choice to use the median to calculate fold changes; meanwhile, the mean value would also be biased, given the fact that the data were not Gaussian-distributed. The authors should be cautious in interpreting their data.

We thank the reviewer for pointing the inaccuracy of our descriptions and have included the parameter used to calculated the percentage and fold increase/decrease in the materials and methods section. Specifically, the mean. Our intention is to plainly state the amount of change seen in a parameter based on the observed changes in the mean value. We agree with the reviewer that interpreting this could be problematic if we are speculating possible mechanisms. Further test should be conducted as to state whether similar increase/decrease changes in a parameter are due to the disturbance of the same mechanisms or different. E.g., we discussed whether the regulation of SYT1 might be or not be the mechanism affected in some of the chimeras that show an increase in the spontaneous release rate, for the release rate observed in some is massively higher than that seen in SYT1-KO (Bouazza-Arostegui et al., 2022). It is tempting to speculate that it could be due to other mechanisms based on the differences in the changes. For this reason, we have given an array of possible mechanisms affected when we manipulate the SNARE domain of STX1.

(7) The authors routinely analyzed the levels of Munc18-1 in neuronal lysates by WB and Munc18-1/VGlut1 by immunofluorescence in various Syx1 mutants. However, in my view, these assays were slightly indirect. It is evident that the SNARE domain of Syx1 participates in the binding to Munc18-1 according to the atomic structures (pdb entries: 3C98 and 7UDB). Meanwhile, Han et al. reported that K46E mutation (located in domain 1 of Munc18-1) strongly impairs Syx1 expression, Syx1-interaction, vesicle docking and secretion (Han et al., 2011, PMID: 21900502). Intriguingly, the residue K46 of Munc18-1, which is close to D231/R232 of Syx1, may have potential electrostatic contacts to D231 and R232 of Syx1. This is reminiscent of the possibility that Syx1D231/R232 and some Syx1-2 chimeras lost their normal function through their defective binding to Munc18-1.nmb, To better understand the underlying mechanism, the authors may need to carry out in vivo and/or in vitro binding analysis between syntaxin mutants/chimeras and Munc18-1. They also need to conduct more discussions about the issue.

We express our gratitude for the identification of a previously overlooked aspect in our investigation of the interplay between Munc18-1 and STX1. In response, we have incorporated additional discourse on this matter in pg11 lines 419-431.

Additionally, we appreciate the thoughtful suggestion regarding additional experiments to further explore the molecular relationship between Munc18-1 and STX1. We agree that co-immunoprecipitation experiments (either by using an antibody against Munc18-1 or STX1 and STX2) would offer greater insight into whether the binding of these proteins is affected in the isoform or the mutants. Notably, we performed immunoprecipitation experiments by using neuronal lysates of the corresponding groups and using STX1A and STX2 antibodies for the pull-downs. However, we were unable to co-IP Munc18-1 when doing so. Changing the conditions of the experiment did not yield better results and so these experiments remained inconclusive for the moment. For this reason, we included it as an open question and a potential concluding hypothesis of the molecular mechanism. However, Shi et al., 2021, have performed co-IP assays using Munc18-1-wt and a mutant form which affects the binding to the C-terminal half of the SNARE domain of STX, and STX1-wt and a STX mutants targeting some of our residues of interest and showed a decrease in the pulled-down levels of Munc18-1 using HeLa cells. We have made sure to mention the conclusion of this important publication in our discussion.

(8) The third possible mechanism (i.e., interaction with Syt1) proposed by the authors seems more reasonable. However, the discussions raised by the authors were not enough. For instance, plenty of literature has indicated that Syt1 may participate in synaptic vesicle priming through stabilizing partially or fully assembled SNARE complex (Li et al., 2017, PMID: 28860966; Bacaj et al., 2015, PMID: 26437117; Mohrmann et al., 2013, PMID: 24005294; Wang et al., 2011; PMID: 22184197; Liu et al., 2009, PMID: 19515907); complexins are also SNARE binding modules that regulate synaptic exocytosis. Lack of complexins could lead to unclasping of spontaneous fusion of synaptic vesicles, though it causes severe Ca2+-triggered release at the same time (Maximov et al., 2009, PMID: 19164751). Meanwhile, different domains of complexin may accomplish different steps of SV fusion, early research had indicated that the C-terminal sequence of complexin is selectively required for clamping of spontaneous fusion and priming but not for Ca2+-triggered release (Kaeser-Woo et al., 2012, PMID: 22357870). Likewise, if possible, the authors may need to carry out in vivo and/or in vitro binding analysis to confirm their hypothesis.

The exploration of complexin´s involvement was limited in our study primarily due to our methodological focus on comprehending molecular mechanisms concerning the sequence disparities between STX1 and STX2. Our laboratory has studied the role of Complexin extensively, and we certainly have had a possible involvement in mind. However, since the sites identified on syntaxin are either conserved between STX1 and STX2 or not close to the central or accessory helical domains of complexin, we did not perform experiments to test putative interactions, and we refrained from discussing complexin in this paper.

(9) Lastly, I would suspect that whether the defects of Syx2 and Syx1 chimeras were caused by the SNARE complex itself, from another point of view that is different from the hypothesis raised by the authors. Changing the outward residues (or we say the solvent-accessible residues) of the SNARE complex may affect the stability, assembly kinetics, and energetics (Wang and Ma, 2022, PMID: 35810329; Zorman et al., 2014, PMID: 25180101), especially for the C-terminal halves. Is this another possible mechanism through which the C-terminus of Syx1 might contribute to SV priming and clamping of spontaneous release? The authors should at least conduct some discussions about the point.

Thank you for this suggestion. We indeed assumed that since the hydrophobic layers of the SNARE domains that form the hydrophobic pocket of STX2 and STX1 are mainly conserved, that the intrinsic stability of the SNARE complex is largely unchanged. Additionally, Li et al., (2022) PMID: 35810329 examined the stability of the alfa-helix structure of the SNARE domain of SNAP25. And while they found no changes in the stability and formation of the alfa-helix when mutating outwards-facing residues for methodological purposes (bimane-tryptophan quenching), their study did not selectively explore the effect of mutations of outer-surface residues on the stability of the alfa-helix.

Zorman et al., (2014) PMID: 25180101, as noted by the reviewer, observed that changes in the sequence of the SNARE domain by using SNARE proteins from different trafficking systems (neuron, GLUT4, yeast…) correlated with changes in the step-wise SNARE complex assembly. However, they also did not selectively mutate the outer solvent-accessible residues, hindering conclusive speculations in the contribution of said residues on the kinetics and energetics of assembly and intrinsic stability of the SNARE complex.

Upon petition of the reviewer, we have added this paragraph to discuss an additional mechanism:

“As a final remark, it is possible that the changes in the spontaneous release rate and the priming stability may stem from a reduced stability of the SNARE complex itself through putative interactions between outer surface residues. Studies of the kinetics of assembly of the SNARE complex which mutate solvent-accessible residues in the C-terminal half of the SNARE domain of SYB2 have shown reduction in the stability of the SNARE complex assembly and are correlated with impaired fusion (Jiao et al., 2018). However, STX1 mutations of outward residues were inconclusive and were always accompanied by hydrophobic layer mutations (Jiao et al., 2018), which affect the assembly kinetics and energetics of the SNARE complex (Ma et al., 2015). Single molecule optical-tweezer studies have focused on the impact of regulatory molecules on the stability of assembly such as Munc18-1 (Ma et al., 2015; Jiao et al., 2018) and complexin (Hao et al., 2023), or on the intrinsic stability of the hydrophobic layers in the step-wise assembly of the SNARE complex (Gao et al., 2012; Ma et al., 2015; Zhang et al., 2017). Although the conserved hydrophobic layers in the SNARE domains of STX1A and STX2 (Figure 1) suggest unchanged zippering and intrinsic stability of the complex, further studies addressing the contribution of surface residues on the stability of the alfa-helix structure of the SNARE domain of STX1 (Li et al., 2022) or the stability of the SNARE complex should be conducted.”

Minor comments:(1) In pg.6, line 236, 'figure 3F', the initial 'f' should be uppercased.(3) On pg.11, line 396, the section title 'The interaction of the C-terminus of de SNARE domain of STX1A with Munc18-1 in the stabilization of the primed pool of vesicles.' The word 'de' is confusing, please check.(4) In pg.12, line 446, the section title, should 'though' be 'through'?

These comments have been acknowledged and changed. Thank you

(2) In pg.7, line 239, '..had an increased PVR (Figure 3G), no change in the release rate (Figure 3I)', should Figure 3I be Figure 3H? and line 240, 'and an increase in short-term depression during 10Hz train stimulation (Figure 3I)', should Figure 3I be Figure 3J? If so, Figure 3I will not be cited in the texts and lack adequate interpretations. Please check.

We apologize for the oversight in not referencing this specific subpanel of the figure and have incorporated the reference in the text. Additionally, our interpretation of this data is connected to the mechanisms that govern efficacy of Ca2+-evoked response, and its dependence on the integrity of the entire-SNARE domain. We wish to highlight the modifications made to the discussion on the regulation of the Ca2+-evoked response based on previous reviewer comment #1, and a similar comment from reviewer #2 (as stated previously).